# Conducting Behavioural Research in the Zoo: A Guide to Ten Important Methods, Concepts and Theories

**Paul E. Rose** [1,2,*] and **Lisa M. Riley** [3,*]

1    Centre for Research in Animal Behaviour, University of Exeter, Exeter EX4 4QG, UK
2    WWT, Slimbridge Wetland Centre, Slimbridge GL2 7BT, UK
3    Centre for Animal Welfare, University of Winchester, Sparkford Road, Winchester SO22 4NR, UK
*    Correspondence: p.rose@exeter.ac.uk (P.E.R.); lisa.riley@winchester.ac.uk (L.M.R.)

**Abstract:** Behavioural research in zoos is commonplace and is used in the diagnosis and treatment of potential husbandry and management challenges. Robust methods that allow valid data collection and analysis constitute an evidence-based approach to animal care. Understanding behaviour is essential to improving animal management, and behavioural research is therefore popular, with a wide choice of behavioural methodologies and theories available. This review outlines ten methodological approaches, concepts or theories essential to zoo science that are based around behavioural observation. This list is not exhaustive but aims to define and describe key areas of consideration when planning and implementing a zoo-based behavioural project. We discuss the application of well-established methods (the construction of ethograms, use of time–activity patterns and measurement of space/enclosure use) as well as evaluating newer or less-widely applied analytical techniques, such as behavioural diversity indices, social networks analysis and Qualitative Behavioural Assessment. We also consider the importance of fundamental research methods, the application of pure science to understand and interpret zoo animal behaviour (with a review of a Tinbergian approach) and consideration of meta-analyses. The integration of observational techniques into experiments that aim to identify the cause and effect of behavioural performance is then explored, and we examine the assimilation of behavioural methods used in studies of environmental enrichment. By systematically studying animal behaviour, we can attempt to understand the welfare of individual animals in captivity, and here we present an example of our reviewed approaches to this area of zoo science. Combining multiple methodologies can lead to a greater understanding of behaviour and welfare, creating robust research, progressing husbandry and advancing conservation strategies. Collaborations between zoological collections and academic researchers (e.g., in Higher Education Institutions) can further refine and enhance the validity of research and husbandry practice alike.

**Keywords:** zoo science; animal behaviour; ethology; observational methods; research methods; behavioural observations; animal welfare

## 1. Introduction

Behaviour is the observable response of individual human or non-human animals to endogenous or exogenous signals [1]. Endogenous stimuli originate internally as part of the animal's physiology (e.g., hormonal changes), and exogenous stimuli are external, originating from the wider environment [2]. Complex and basic animals alike survive by continuously, and often instinctively, reacting to stimuli, culminating in behavioural expression with adaptive value [1]. This alters the phenotype of (sub)species, leading to individual differences within species and populations [3]. Behaviour includes the large-scale actions of an animal, such as locomotion, feeding and social activity, but also the gait, posture, gestures, facial expressions, vocalisations and other communicative signals an animal may express [1,4]. The fact that behaviour is obvious, measurable and can be manipulated makes it one of the biological scientist's most useful tools for investigating the natural world [1]. Animal

behaviour studies typically, both historically [5] and currently [6], comprise a large proportion of zoo and aquarium (hereafter "zoo") research output, potentially because of both the ease of recording such data and the many ways behavioural observations can be integrated into wider research projects.

In zoos, behaviour is essential to inform an evidence-based approach [7] to animal husbandry and population management. Determining how to care for captive animals using research to inform decision-making is essential if we are to allow captive wild animals to thrive rather than merely survive in the zoo [8–10]. By first observing behaviour, and secondly attempting to interpret meaning, we can compare captive and wild behaviour, understand animal behavioural needs, cognition and preferences, increase reproductive success, change enclosures to allow a greater repertoire of behavioural expression, understand what aspects of enclosure provision and wider husbandry are important to the animal and minimise the influence of captive stressors [10–12]. The remit of zoo research has expanded in scope and type over the years, but behaviour remains a consistent focus and methodological application [6,13].

Animal behaviour (in pure and applied contexts) is commonly taught as a stand-alone degree in the university sector (from Foundation to Masters' degree) as well as part of broader degree topics, such as Psychology, Conservation Biology or Animal Science. Likewise, a proliferation in zoo-specific qualifications (e.g., Zoo Biology, Zoo Conservation Biology and Zoo Animal Welfare), from college diplomas to postgraduate degrees, is noted [14], reflecting a student-led demand to study zoo animal management as a core topic of further and higher education. Students and those seeking employment in the animal management or conservation industries need to understand and be able to apply behavioural methodologies and key theories.

All behaviour-based projects, regardless of the specific methodology, start with a broad research question. Planning of the study radiates from this central point with some almost universal considerations common to nearly all research (Figure 1). Any aims need to be defined to guide the planning of methods and provide a rationale for the research. For most quantitative and some qualitative approaches, hypotheses will be constructed to direct what data are collected, how data are collected and the data analysis process. For zoo-based behavioural research, questions often arise based on certain species, commonly due to a lack of evidence for the care of that species in the zoo or a desire to know more about the species' responses to husbandry and management. For basic science questions, species selection may be based on the most appropriate candidate for answering that specific question (for example, when the aim of the study is wide, or one is conducting an over-arching investigation of fundamental principles, such as behavioural plasticity or adaptation to captivity).

Understanding the natural ecology of the species is important when considering what to measure and when; therefore, an ethogram is required to define behaviours. The focus must then shift to considering behavioural sampling (what, when, how and whom to collect behavioural data from). Reviewing available information on temporal or seasonal activity patterns will help with deciding when/where to place trail cameras (or similar technology for remote monitoring) or observe in person to provide the best chance of capturing maximal behavioural data, or providing opportunities for behavioural manipulation (such as enrichment) to maximise the subject's engagement. The species' natural history and behavioural ecology will also influence the timeframe of data collection and how long needs to be spent recording behaviour, which also needs to account for the zoo's hours of opening (i.e., access to the sample population and/or placement and access of remote monitoring equipment). Answering these questions will ultimately allow specific behavioural techniques to be selected and applied for successful data collection. Which behavioural methodological approach or theory is most helpful for which specific question?

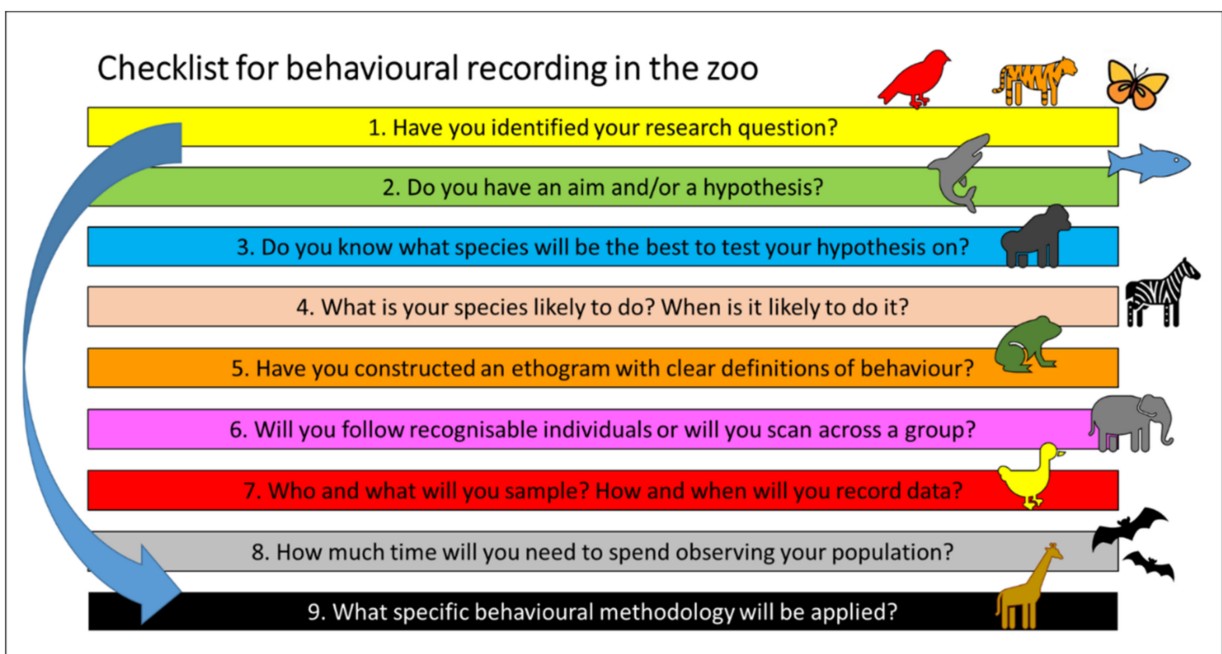

**Figure 1.** Flow diagram for how to plan and construct the initial experimental design for behavioural data collection in the zoo and aquarium. Once it is known how individuals will be selected for study and how long will be spent on behavioural observations, planning of the statistical analysis based on the types of data generated from these observations is possible.

The aims of this paper are to provide an explanation and description of important methodological approaches and key scientific concepts and theories to the study of animal behaviour in the zoo, to guide the end stages of methodological planning and the collection of valid and relevant data. We explain key terms and methodological principles and direct the reader to useful sources of further information should this method be most suited to the development of their behavioural research project. This paper is primarily aimed at students and new-to-the-zoo researchers but may also be of help to more established scientists looking to diversify their application of behavioural techniques. Given the lack of repeatability of methods and reproducibility of results noted in reviews of published scientific studies [15,16], this paper aims to introduce the fundamentals of ten behavioural methods or concepts applicable to zoo studies so that any researcher, regardless of level, can commence data collection with confidence in the validity and robustness of their chosen approach.

## 2. Behavioural Methods, Concepts and Theories

In this section, we provide definitions and explanations of ten behavioural methods (ethograms, time–activity budgets, space/enclosure usage calculations, behavioural diversity indices, social network analysis, qualitative behavioural assessment, meta-analyses), concepts (experimental studies and repeated measures, assessing enrichment using behavioural data) and theories (Tinbergen's Four Questions) to help the zoo researcher decide on their best approach when developing a research design for observation data collection. As with all scientific research, it is essential to have the methods for data collection ethically reviewed and scrutinized before data collection commences.

### 2.1. Developing an Ethogram

An ethogram is a list of behaviours and associated definitions [4] that provides a way of categorising and thus differentiating individual behaviours within the repertoire of an individual or species. Each behaviour must be discrete with mutually exclusive definitions based on the mechanical actions that are observed rather than on any intentionality, motivating the expression of that behaviour. For example, "feeding" behaviour may be defined as "*the processing, mastication and swallowing of food items using the mouth*

*and tongue*" rather than "*the animal is eating food*". The latter definition does not explain the mechanistic processes of feeding that the researcher will observe nor does it capture the potential motivation behind the behaviour's performance. The ethogram should be comprehensive and meaningfully represent the behavioural repertoire of the animal [17]. For many behavioural studies in the zoo, a simple ethogram that focuses on key state behaviours or behavioural categories [18,19] needed at a specific time or for a specific focal species or population is required. Ethograms often contain an "other" and an "out of sight" category to enable recording of unusual behaviour and time when the animal was not seen. An example of such an ethogram is provided in Figure 2, taken from [20]. Similarly, easy-to-follow definitions of behavioural states from ethograms of a longitudinal study [21] and research using remote technology [22] are available in the literature.

| Behavioural state | Description |
|---|---|
| Walk /run | Bipedal movement along the ground either at a slow or hurried pace. Running birds may have outstretched wings. |
| Rest /sleep | Motionless with head "tucked under wing" standing or sitting, with eye(s) open or closed. |
| Preen / bathe | Cleaning and oiling feathers with bill. Or using water to wash feathers by scooping water over body with wings and/or bill. |
| Feed / forage | Consumption of food from feed trough or natural filtering (pumping water through bill) in pools. |
| Stand | Motionless. Not alert (head is held low in front of body), not asleep or resting. General inactivity. |
| Alert | Neck held in erect S-shape with head on 90° angle, scanning surroundings. |
| Social | Long-duration positive social association defined as the following of one bird by another around the enclosure. |
| Courtship / nesting | Courtship: Long duration head-flagging (movement from side-to-side) or marching displays, or extended wing saluting (spreading of wings out to the bird's sides). Nesting: Nest mounds constructed using bill to gather damp substrate together. |
| **Behavioural event** | **Description** |
| Vocal | Producing a range of calls (grunting and honking noises). |
| Following | In pairs or trios. When one flamingo moves, others mirror its action. |

**Figure 2.** Example of an ethogram from published research on two species of flamingo used to categorise and record the performance of state (long term) and event (short term) behaviours. Taken from Rose [20].

The aim of an ethogram is to standardise data collection between sample periods, observers, animal populations and different studies so that results are directly comparable and can collectively add to our understanding of animal behaviour. The development of an ethogram is a fundamental preliminary step to nearly all other behavioural methodologies or applications, such as the construction of activity budgets and the evaluation of environmental enrichment. The ethogram allows repeatability as others can use or adapt a previous ethogram with successive edits over successive research projects. An early step in

the research process is to source information for an ethogram from the literature and to then test these behavioural definitions via preliminary observations. This enables the evaluation of the ethogram to adequately describing what has been observed. Observers must continue to review the ethogram throughout data collection to ensure that categorisation of behaviour remains consistent.

Numerous ethograms for zoo animal behaviour exist in the literature, including useful examples in Stanton et al. [23] and Smith and Wassmer [24]. Published behavioural inventories, e.g., the list of behaviours performed by giraffe, *Giraffa camelopardalis* [25], are excellent tools to help guide preliminary investigations into what behaviours are likely to be seen in the zoo and how they could be defined and categorised. The researcher can select an ethogram from published research to adapt and compile for their own work but must be mindful of the usefulness and quality of available information. Never assume that a published ethogram can be used in its entirety without prior review or adaption to the specifics of the animals or system being used for data collection.

To ensure that the ethogram is useful in the field and allows for the appropriate identification of behaviours for data collection, intra- and interobserver reliability should be calculated from a sample of observations during a pilot study to ensure the validity of the final dataset [26]. Intraobserver reliability allows a single observer to check that they are consistent in how they are applying their ethogram to categorise behaviour from their observations. Interobserver reliability is the same process but checks that a group of observers are following the categorisation of behaviour in the same way. Both calculate a percentage of the number of correct identifications of behaviour compared to the overall number of identifications conducted. Further explanation of reliability testing, including the application of Kappa testing to reliability scores, is provided in Kaufman and Rosenthal [26], and the formula for percentage calculations can be found in Bateson and Martin [4].

An ethogram must be consistent (in its definitions and descriptions of behaviour) within a study, but they can be "working documents" across studies. As further scientific discovery is made, further behaviours are added to the ethogram, and previous definitions are reviewed. Ethograms can become extensive based on years of systematic observation of behaviour. For example, an audio-visual encyclopaedia of wild chimpanzee (*Pan troglodytes*) behaviour by Nishida et al. [27] assimilated over 60 years of chimpanzee research and documented nearly 200 pages of behaviour definitions/examples derived from over 300 publications.

### 2.2. Creating Time–Activity Budgets

Time–activity budgets are standardised graphical summaries of behavioural expression, where each individual behaviour (e.g., climbing) or behavioural category (e.g., locomotion—climbing plus running, plus leaping, plus walking, etc.) is expressed in percentage of total time or occurrences of all observed behaviour. The qualitative definitions of behaviour from the ethogram are quantified into the total amount of time an individual or population spends performing a named behaviour, expressed as a proportion/percentage of total behaviour. To conduct this, animals must be observed repeatedly over time and behaviour systematically sampled and recorded. Numerous texts expand on the sampling and recording methods useful to observational study [4,28]. Sampling includes focal follows of an identifiable individual or scans across a group, and recording can be instantaneous recording on pre-determined time points or continuous recording of each behaviour as it is performed by the animal [4]. It is important to remember that when scan sampling one animal group in one zoo, the averaging of behavioural data across the individuals will result in a sample size of n = 1. Consideration of this n = 1 is needed to guide appropriate statistical testing as well as evaluation and application of the study's findings.

Having developed an ethogram, the behaviour needs to be broadly categorised as a "state" (long duration behaviours) or an "event" (short, momentary actions or re-

sponses) [4], as generally, only state behaviours can be displayed as a proportion of the animal's time–activity budget. Event behaviours, counted during the sampling period and displayed as a frequency or rate of occurrence, can help explain the reason why animals are dividing their time between specific states. For example, changes to homeostatic foraging (state) and maintenance (state) behaviours of animals in the breeding season due to more energy being devoted to courtship actions (events). The species being investigated, the timeframe available to the researcher and the question being asked will influence the choice of sampling and recording method. Continuous recording of a focal individual that performs two or three infrequently changing states will be easier to conduct than in an individual with a varied behavioural repertoire with short amounts of time spent per action. In this case, an instantaneous sample may be more useful. As illustrated by Figure 3, each different sampling and recording method may yield a slightly different time–activity budget, and this should be considered by the researcher during the planning and pilot study stage.

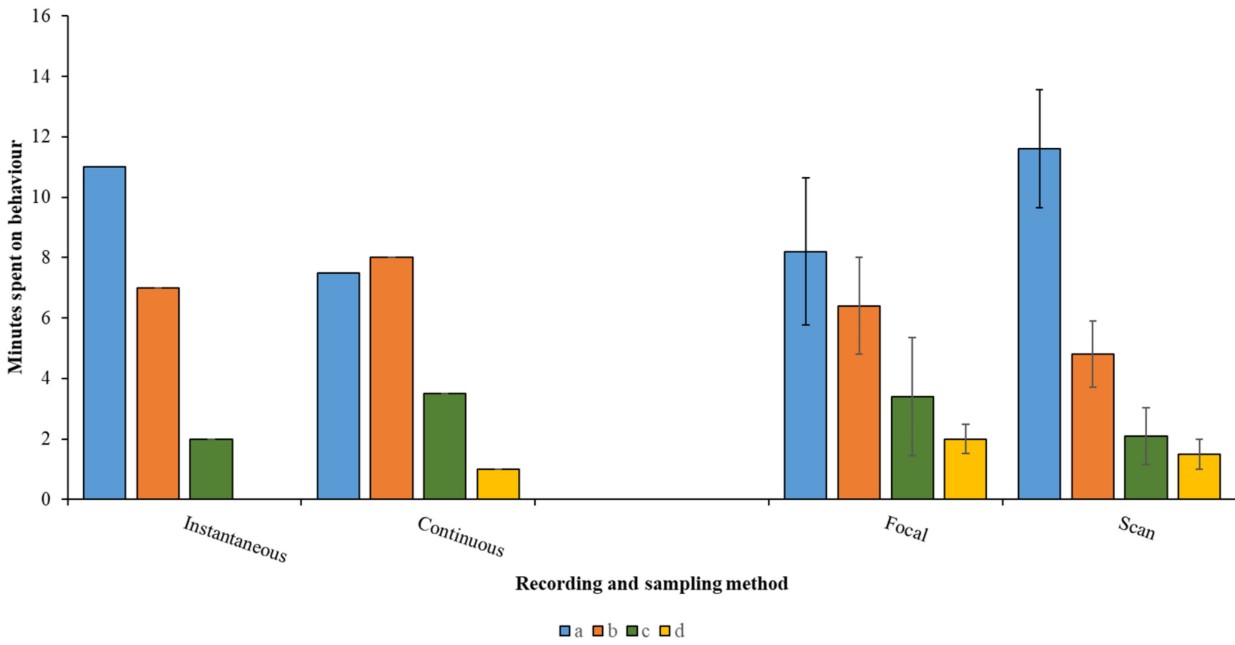

**Figure 3.** Example data for four behaviours recorded for 20 min using two common recording and sampling methods. a–d are example behaviours. Instantaneous recording misses behaviour d completely and suggests more time is spent on behaviour a, whereas continuous recording provides data on all four behaviours and suggests equal time spent on behaviours a and b. Focal sampling (following an individual animal) provides the same overall pattern across all four behaviours as for scan sampling (counting the number of animals performing behaviour in a group), but scan sampling gives a larger difference between behaviours a and b. "Eyeballing" these data and the overlapping error bars suggest that even though differences in the proportion of time allocated to each behaviour vary for focal and scan sampling, such difference may not be significant.

Time–activity budgets are exceptionally useful pieces of research output that allow inference of underlying motivational states, and the importance of behavioural needs to be discussed. They can provide researchers and zoo practitioners with helpful information on what the animal needs from its environment. Therefore, a review of a time–activity budget can lead to alteration in management, perhaps to reduce the proportion of time spent on abnormal behaviours or to promote behaviour patterns conducive to successful reproduction. For example, conservation of behaviour (i.e., to ensure that adaptive behaviour patterns remain in captive populations) or behavioural culture (i.e., the unique behavioural responses of a specific population from a specific geographical area [29]) can be better undertaken if time–activity budgets are estimated and evaluated. The resulting data can evidence the need for alterations to husbandry and management to promote

specific behavioural outputs for all species (but especially those species who are part of conservation action planning).

Along with consistent use of the same ethogram across studies, quantitatively summarising behaviour in a time–activity budget that expresses behaviour as a function of overall time, rather than just the total number of minutes spent performing a behaviour, means direct comparisons between studies can be made even if the total time animals were observed in each study was different. In turn, this provides an opportunity to pool data across studies and across decades of research.

*2.3. Measuring Space and Enclosure Usage*

As zoo animals live in controlled conditions where their immediate environment is created for them, it is important to assess how they use the space they are provided with as both the amount of space and quality of space will impact the performance of behaviour. Combining behavioural measurement to create time–activity budgets with enclosure usage enables evaluation of activity patterns in the context of the space available for the individual and/or group of animals. A common way of assessing enclosure usage is to divide an enclosure into a number of zones of equal area and use the spread of participation index (SPI) to calculate any difference in the observed occupancy time compared to the expected frequency of occupation (equal use of all zones) [30]. This is achieved by simply recording the zone that is used every time behaviour is recorded.

It is important to consider that zones within an enclosure may hold different levels of value for the occupants. For example, small but highly valued zones may be used more than larger, less ecologically important areas [31]. Not all zoo enclosures can be divided into equally sized zones. Given this and differential resource value, the modified SPI formula should be used, which compensates for unequal zone sizes [30]. The results generated from both SPI equations should be considered descriptive statistics, with a result of 0 suggesting equal occupancy of all zones and a result of 1 identifying highly uneven zone occupancy. Further inferential analysis of SPI values can be used to identify potential predictors of zone occupancy, e.g., environmental, social, anthropogenic or husbandry influences [32].

Enclosure usage can also be estimated using an electivity index (EI) which calculates an animal's actual utilisation of specific features (e.g., resources within the habitat) of the environment in relation to the overall availability of these features. The use of EIs across a range of ecological constructs is reviewed in Lechowicz [33], and EIs were used to specifically measure enclosure usage preferences in the zoo in papers on gorillas (*Gorilla gorilla gorilla*) and chimpanzees [34], African wild dogs, *Lycaon pictus* [35] and Round Island geckos, *Phelsuma guentheri* [36]. EI allows estimation of preferred features of the enclosure based on those that are limited in availability (e.g., small but valuable area) but are overused by the animals given the proportion of availability of this feature compared to the total space available in the enclosure. Usage of these preferred features can achieve a maximum score of +1. Areas of the enclosure that are occupied less than they are proportionally available within the enclosure overall would score maximally −1 and would be considered avoided features. An enclosure feature can be scored a 0 if used randomly (i.e., in a non-preferential or non-avoided way), and this suggests the most equal usage of this feature in the enclosure (based on the availability of this feature and the animal's utilisation of it).

By considering SPI or EI index scores in combination with the number of times each zone or resource was used, SPI and EI can identify the zones/resources that animals utilise and seek out, and therefore reveal information about behavioural needs and preferences. If individuals within a group are recognisable, data from focal follows can allow calculation of individual EIs or SPIs; therefore, any monopolisation of a valued resource by one individual to the detriment of others can be further studied if identified. Public presence, social dynamic or other pressures from the environment could then be investigated, manipulations made to the animal's environment and then enclosure usage observed and recalculated to determine any change to SPI or EI values. The formulas for each of these

space usage methods are provided in Table 1, alongside references that explain the theory and that have used the theory to analyse enclosure usage data.

**Table 1.** Examples of three commonly used formulae for calculating enclosure usage via observational data collection.

| Spread of Participation Index [37–39] | Modified Spread of Participation Index [30,40,41] | Electivity Index [42–44] |
|---|---|---|
| Formula $M(n_b - n_a) + (F_a - F_b)/2(N - M)$ $M$ = mean frequency of observations per zone $n_b$ = number of zones with observations < M $n_a$ = number of zones with observations > M $F_a$ = total observations in all zones with observations < M $F_b$ = total observations in all zones with observations > M $N$ = total number of observations in all zones | Formula $\sum \lvert f_o - f_e \rvert /2(N - f_{e\,min})$ $f_o$ = observed frequency within a zone $f_e$ = expected frequency within a zone $N$ = total number of observations $F_{e\,min}$ = expected frequency within the smallest zone | Formula $W_i - (1/n)/W_i + (1/n)$ $W_i$ is calculated as: $r_i/p_i)/\sum r_i/p_i$ $r_i$ = observed use (e.g., amount of time) of resource i $p_i$ = expected use of resource i $n$ = number of different resources |
| Application Exhibits that are uniform in size and structure where resources are evenly spaced. | Application Most zoo enclosures where resources are uneven in size and distribution, and the overall shape of the enclosure is irregular. | Application Evaluation of new enclosures and post-occupancy evaluation. Calculation of individual animal preferences for zone occupancy. |

The validity of zone/resource use data must be considered. What constitutes a zone within an enclosure is particularly relevant to validity considerations. Zones need to be based on their ecological relevance and value to the animal. Differences in habitat structure and quality can be useful starting points for assigning zones (e.g., deep pool, shallow pool, mature trees, open grassland), but too many zones can impact the quality of the final enclosure usage measurement. Zoning a large enclosure into many small zones can artificially inflate the chances of finding unequal, biased enclosure usage when using the modified SPI [21,32]. Google Earth Pro [45] is useful for mapping out enclosure zones in medium to large open enclosures where boundaries are known and not obscured by buildings or vegetation. Google Earth Pro's in-built "draw polygon" function allows for defined zones to have their surface area mapped, which can then be used for calculating expected frequencies of enclosure zone occupancy.

*2.4. Calculating Behavioural Diversity*

The calculation of behavioural diversity can be an important way of determining how husbandry and management affect the potential range of behaviours displayed by an individual or species group [21,46]. A comparison of behavioural diversity indices (BDIs) from species populations across different zoos can identify where and for whom scores are relatively higher, and therefore where/when increased behavioural diversity is observed, and why this may occur [47]. This further provides evidence for best practice guidelines to implement husbandry and management practices that enable a more complete behavioural repertoire to be expressed. Research into the natural behaviour patterns and behavioural ecology of the species (used to construct the study's ethogram) prior to data collection is essential to determine those behaviours that are biologically relevant. Behavioural diversity calculation should include only normal or natural behaviours that typically provide adaptive or fitness benefits. The behavioural scientist needs to remember that such adaptive or fitness benefits do not have to be obvious to the observer. There may be an underlying motivation to perform a behaviour that is currently not understood. An animal that performs a large number of different abnormal repetitive behaviours may generate a high BDI, but this would be inappropriate as a measure of behavioural normality or husbandry relevance for that particular individual. Thus the researcher should familiarise themselves with the normal behavioural repertoire for the species and be mindful to include only those behaviours in any calculations. Abnormal behaviour, though excluded from the BDI calculation, must be discussed and evaluated given that its performance is an artefact

of captivity and provides data on animal welfare states. BDI formulae from the ecological sciences (used to calculate species diversity within a habitat) can be easily applied to behavioural research data from zoo populations. Common methods used include the 1-Simpson's Index [48] and the Shannon H-Index [49]; the formula for each method is provided in Table 2.

**Table 2.** The formula for the Simpson's and Shannon ecological diversity indices as adapted for animal behaviour.

| 1-Simpson's Index | Shannon H-Index |
|---|---|
| $1 - \sum n_i(n_i - 1)/N(N - 1)$ <br><br> $n_i$ = the total time spent on a particular behaviour. <br> N = the total time spent on all behaviours. | $H = -\sum(p_i \ln p_i)$ <br> $p_i$ = the relative abundance of each functional category, calculated as the proportion of behavioural elements of a given functional category to the total number of behavioural elements of all functional categories ($n_i/N$). |

BDI may be especially helpful when determining the effects of enrichment on natural behaviour performance as data gathered for calculation of time–activity budgets can be used to determine BDI pre-, during- and post-provision of enrichment. Likewise, the use of space and behavioural diversity may provide relevant insights into how animals split their time between different behaviours and different enclosure areas. Modification of zones based on their occupancy and value to the animal could promote behavioural diversity if animals are inactive or lethargic. Cronin and Ross [50] discussed why care should be taken when using BDI to prevent inappropriate conclusions from being drawn (e.g., if the BDI is inaccurately calculated or its limitations not considered before use) that may erroneously inform husbandry and welfare needs.

*2.5. Social Network Analysis*

It is important to measure the effect of the social environment, specifically for species that gain adaptive or fitness benefits from the social relationships they form and the social dynamic they live within. Social network analysis allows the fine-scale structure and function of a social group to be thoroughly investigated, with individuals in the network represented as nodes that are connected together by edges (the relationships that exist between them) [51]. Data can be collected as associations (proximity measures, akin to the animal's behavioural state) or interactions (a response from one individual based on the action of another, akin to an event behaviour). The basis for the inferred social relationship must be a biologically relevant metric for that specific species, e.g., nearest neighbour proximity using neck length for giraffes [52] or grooming interactions for identifying chimpanzee social bonds [53]. Social network data can be permuted, and null models created to evaluate the effect of structure and stability of the network, essentially asking "*is the observed patterning of associations/interactions significantly different from what would occur if all social ties were random?*" Temporal effects on relationships, to provide an understanding of how long social ties may last, can be run using lagged association rates (LAR) [54]. The LAR is the probability that two individuals are in a social relationship given their association some time lag earlier [55]. Measures of centrality (i.e., inferences of the most important or influential members of the network) can also be calculated to show how influential certain individuals may be within the social group [56].

More information on the application of social networks to the zoo and what types of questions can be answered with a social network analysis approach is provided in Rose and Croft [57]. Information on how social network analysis is relevant to applied ethological study generally is explained by Makagon et al. [58] and for conservation specifically by Snijders et al. [59]. The application of social network analysis to the measurement and assessment of animal welfare is discussed by Beisner and McCowan [60]. How to generate questions, design methods and use network software to test hypotheses are covered in a useful guide by Croft et al. [61], and the application of null models for inferential testing (e.g., compare association patterns if a network's structure was random compared what has been observed) is explained in detail by Farine [62].

Social networks research relies on individuals being recognisable and identifiable when followed for data collection. Information present in zoo animal records can be useful for including in network analyses as attribute information (i.e., individual characteristics) that can ultimately help decipher the reasons behind specific preferred or avoided social relationships. Social networks collected over different years or seasons can be compared against each other (providing that the methods used are standardised and the approach to observations has not changed) to determine longer-term patterns of sociality that may provide information on how animal's respond to the managed social environment of the zoo. Packages in the R licence-free statistical software [63] suitable for networks analyses, such as "asnipe" [64], are outlined in Farine and Whitehead [65], and stand-alone networks software, e.g., "Socprog" [66] and "Netdraw" [67] are also available for free download too.

### 2.6. Qualitative Behavioural Assessment

Qualitative Behavioural Assessment (QBA) aims to capture how an animal's body language and the qualitative way in which the animal performs behaviour illustrates the underlying emotional state [68]. Here, instead of focusing on what behaviours the animal is performing, the observer uses behavioural descriptors of emotions and considers the extent to which each descriptor applies to that animal. This is a "whole animal" approach (the animal's state of being is holistically considered) which means the dynamic way in which the animal interacts with its environment can be captured in a repeatable manner [69]. Descriptors of the animal's emotions are not based on the mechanics of behaviour as would be used for an ethogram, but rather are terms that define and explain the underlying feeling of the animal (for example, relaxed or agitated, comfortable or restless). Descriptors can be produced by free choice profiling (adapted for each individual) or fixed lists (universally applied for the species and scenario) [70] and scored on a visual analogue scale from 0 (not describing the emotional expression of the animal at all) to 125 mm (completely describing the animal's emotional expression). Free choice profiling allows multiple observers to generate their own list of terms, and after the animal has been watched, analytical techniques (such as Generalised Procrustes Analysis—see Wemelsfelder et al. [71]) are applied to these data to reduce all descriptors to two or three key explanations. QBA conducted using a fixed list approach sees observers being provided with a list of pre-determined descriptors, trained to observe these behavioural expressions, and then data pooled for analysis using a principal components analysis (PCA) or similar (to group similar descriptors together for ease of interpretation). For this reason, reliability scoring of all those who collect data for QBA is essential to ensuring that data are valid and methods followed accurately [70]. As QBA is relatively novel and sometimes criticised for being subjective, many researchers tend to correlate QBA scores (or the PCA scores associated with QBA scores) with traditional behavioural sampling data (such as time–activity budget data) or related physiological measures, which can, if significant correlations are achieved, suggest that construct validity is achieved.

Examples of QBA-focussed research pertinent to zoo animal welfare are reviewed in Rose and Riley [9], which provides researchers with examples of the practicalities of QBA data collection and how evidence can be collected for husbandry changes. QBA was successfully utilised to understand responses to anthropogenic stressors in the zoo, such as the presence of zoo visitors [72] and involvement in direct human interactions [73]. Species-specific welfare assessment tools that feature QBA for zoo animals were also published, including a tool for elephants (*Loxodonta africana*, *Elephus maximus*) [74] and for waterfowl (Anseriformes) [75], showing the growing importance of QBA as a methodology for informing on individual and group-level welfare states.

### 2.7. Meta-Analyses of the Behavioural Literature

A meta-analysis is a statistical tool designed to determine mean (and variance) population effects (statistical relationships) and therefore evaluate the validity of data from multiple studies that ask the same (or similar) research question(s) [76]. Meta-analyses

offer a great opportunity to synthesise new information from previously published studies to answer a novel research question and are easily applicable to the wealth of zoo-based behavioural studies. This approach is especially useful to the gathering of evidence on natural behaviour patterns and wild time–activity budgets where a systematic review of published papers and more formal meta-analyses can help to provide the baseline information for evaluation of the behavioural repertoire performed by a species in captivity. Meta-analyses combine the results of multiple, previous studies to generate novel information; examples of such an approach to behavioural research can be found in Wood et al. [77], investigating the species differences in waterbird aggression, and in Shyne [78], who identified species-specific needs for environmental enrichment to reduce abnormal behaviour performance in zoo-housed mammals. Seminal research by Clubb and Mason [79] evidences how data from meta-analyses can have practical applications to the zoo by examining how husbandry practices for large carnivores should be modified based on behavioural evidence from wild research. Structured literature reviews help create and refine research questions, for example, allowing us to judge ecological drivers of adaptive behaviours, e.g., social grouping, vigilance, foraging strategy and habitat choice [80], to enable the promotion of these behaviours (and recreation of the functional units of the species' habitat) in the zoo. Help with standardised methods for meta-analyses and structured literature reviews is available; for example, the Preferred Reporting Items for Systematic Reviews and Meta-Analyses (PRISMA) approach is widely utilised and easy to follow [81].

### 2.8. Applying Tinbergen's Four Questions

Tinbergen's Four Questions (two of proximate and two of ultimate consequence) were first published in 1963 to help scientists decipher the reason behind the performance of a specific behaviour, how it may develop over the course of the individual's lifetime and what benefits it has to the population/species overall [82]. The application of a Tinbergian review on behaviour of zoo-housed animals involves the assimilation of previous research or conducting new research to determine: Causation/Control of behaviour—the physiological mechanism that leads to behavioural expression; Ontogeny/Development of behaviour—how the behaviour is acquired within the lifetime of the individual, e.g., learning, maturation; Function/Adaption of behaviour—how the behaviour is linked to survival of the individual; Evolution/phylogeny—why the behaviour has persisted across generations. Directed research into the performance of abnormal behaviour in zoo-housed animals can be based on the interpretation of behavioural needs and the ecological importance of actions that the animal performs. This allows evaluation of why behavioural problems occur when behaviour is constrained by the captive environment, and how we can promote the adaptive function of behaviour (i.e., promoting the fitness benefit that an individual achieves from a behaviour) by altering this environment to make it more ecologically relevant.

Mellor et al. [83] expand on how to use a Tinbergian review of abnormal repetitive behaviour in captive birds from a range of species to provide a deeper understanding of why such behaviours occur (using the wild ecology of the animal as the starting point). These authors highlight the importance of the developmental stages of behaviour to influencing the performance of abnormal activities later in life. Expanding the Tinbergian approach to more zoo behavioural research questions could shed light on husbandry challenges as well as promote beneficial husbandry changes that are supported by evidence on what the animal needs to do from an evolutionary and ontological perspective. One potential application is to perform a Tinbergian review on positive or negative behavioural welfare indicators. For a positive indicator, assessing the effects of a lack of performance of that behaviour on animal welfare, using supporting knowledge of behavioural ecology and direct data collection can improve husbandry and enclosure design. Olsson and Keeling [84] used Tinbergen's Four Questions to evaluate dust bathing (a behavioural indicator of positive welfare) in junglefowl (*Gallus gallus*) and domestic poultry (*G.g. domesticus*) but also note the importance of the early development period on future behavioural preferences and

performance. This type of application of the Tinbergian methodology is relevant for all zoo animals. Table 3 explains the Tinbergian approach to welfare study using browsing in giraffes as the example behaviour to illustrate how behavioural research using the Four Questions could help provide answers to welfare-related issues.

**Table 3.** Tinbergen's Four Questions applied to rumination behaviour in the giraffe. Extrapolation of this approach to all ungulates can allow for assessment and explanation of poor welfare and evidence the most suitable ways for changing practice. Underlined text is the name of the Tinbergian Question, plain text explains the behaviour in terms of the question, italicised text evaluates the potential welfare impact of not allowing the behaviour to proceed as normal.

| Proximate | Ultimate |
|---|---|
| Causation/Control | Function/Adaptation |
| Browsing is instigated by hormonal signals relating to hunger and well as visual cues that identify suitable foraging opportunities (e.g., colour, smell, taste of suitable leaves). | To provide highly fibrous material for internal, ruminal microorganisms to break down, producing volatile fatty acids for metabolic functioning. |
| *Without regular opportunities for browsing, stereotypic chewing and mouthing behaviours, and locomotory pacing will occur.* *Homeostatic behaviour performed in response to fibrous food present in the reticulum and rumen. Ingestion of forage material and processing of food in the mouth will illicit rumination activity.* | *Lack of opportunities to forage and ingest fibrous material disrupts the environment of the rumen, changing the colonies of microbes that can cause ruminal acidosis and other health concerns. Assimilation of energy and nutrients is negatively affected.* |
| Ontogeny/Development | Evolution/Phylogeny |
| Sex differences are noted in foraging behaviour in the giraffe, with males giraffes commonly browsing at full height and females feeding at lower levels. Young giraffe begin to browse at 3–4 months of age and travel with their mother and other females to suitable browsing sites at 6 months of age. | No mammalian species can digest cellulose. Ruminant herbivores have a symbiotic relationship with rumen microbes to enable the production of energy from cellulose. Browsing behaviour is highly selective, meaning the giraffe can forage optimally within its habitat and ensure a constant supply of cellulose to these microbes. |
| *Limited opportunities to express differences in foraging behaviour in the zoo may results in poorly defined and incomplete behavioural repertoires, which may be unfulfilling to the animal.* | *Change to the microbiota of the gut occurs in captivity when substitute diets are fed. The health of these gut bacteria impact on health and wellbeing of the animals. Change to the natural flora of the gut due to zoo diets may have long term impacts on population health and sustainability.* |

## 2.9. Experimental Studies and Repeated Measures

Behavioural change can be an important dependent (measured) variable when observational methods are incorporated into experimental studies. The experimental design attempts to not simply record behaviour (as observational studies do) but to elucidate a causes/effect relationship. What causes behaviour change when a treatment is experienced by the animal(s)? In the zoo, treatment could be a change to diet or food presentation, access to enrichment or to another part of an enclosure or member of the social group or use of operant condition training or other change in husbandry or daily routine. The treatment is manipulated and is, therefore, the independent variable, whereas the animal's behaviour is consistently the dependent variable. Experiments, so-called because the researcher exerts "control" of the independent variable by manipulating it, can use an "independent groups" or "repeated measures" design. An independent groups design, where the behaviour of different groups of animals is compared, is of more limited use in behavioural research (with the notable exception of wild vs. captive group comparisons) as individual animals behave independently, even when part of a group. A repeated measures design is typically of far greater use and involves the same animals being observed repeatedly, before (baseline) and after treatment (change in the independent variable). Individual variables (such as variation in the characteristics between animals, e.g., personality, cognitive bias) are not an issue under a repeated measures design, allowing genuine behavioural responses to be measured.

Order effects are problematic in a repeated measures design as the order in which the change in the independent variable is experienced could cause erroneous behavioural responses as an artefact of testing, wrongly attributed to a change in independent variable.

For example, if testing the change in performance of abnormal behaviour in response to the installation of privacy screens, and the animal significantly reduced time spent stereotypically pacing after privacy screens were added, was this a genuine effect of the privacy screens or an artefact of now being familiar with the testing paradigm? Or was some extraneous variable, such as a change in weather (something beyond the researcher's control), the cause of behaviour change across the two time points? One can control for order effects by counterbalancing the experimental design (Figure 4), making the interpretations of repeated measures studies more consistent and more valid.

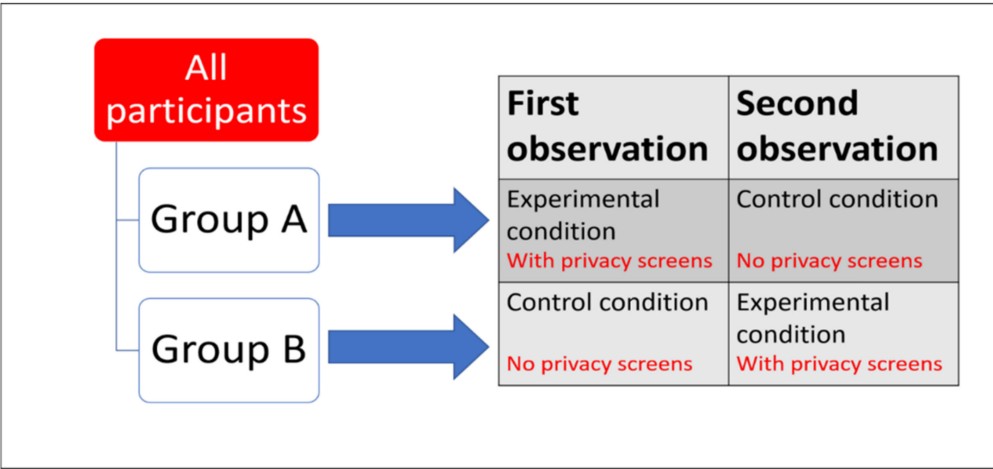

**Figure 4.** Example of a counterbalanced repeated measures experimental design. Dependent Variable = time spent stereotypically pacing. Independent Variable = introduction of privacy screens. All animals experience both conditions (no privacy screens vs. privacy screens present). If time spent stereotypically pacing was significantly reduced in the experimental condition compared to the control condition, results are very unlikely to be due to order effect and are more likely to be due to the treatment effect.

There is a suite of parametric and non-parametric inferential tests and numerous statistical software packages that exist specifically to deal with repeated measures data, for example, the "lmerTest" package for the R statistical analysis programme [85] that runs repeated measures mixed models on large and complex datasets. Identification of fixed and random factors and covariates is crucial to correct interpretation from such modelling of behavioural data. If the levels of a factor can be controlled, then the factor is fixed in the model (for example, presence or absence of enrichment). If the levels of a factor from within the population were sampled randomly, then the factor is random (for example date that observations were conducted on). As data points are not independent across the conditions (i.e., two data points are from the same animal), it is essential to use repeated measures inferential tests to maintain the validity of the research.

### 2.10. Evaluating Environmental Enrichment

The use of enrichment is commonplace in zoos and involves the provision of equipment, objects, social agents or sensory items to promote naturalistic behaviour patterns or expand the behavioural repertoire of captive animals. The application of enrichment can be made more impactful by evaluation of how longer-term, positive effects on behaviour patterns are enabled. Whilst numerous categories of enrichment are noted, for example, nutritional, occupational, physical sensory and social [86], the action of enrichment is continuous and not mutually exclusive. For example, the provision of a scatter feed may constitute a form of nutritional enrichment whilst also providing opportunities for occupational, physical, sensory and social enrichment.

The use of enrichment should promote a diverse range of beneficial behavioural responses in the animal and provide opportunities for a positive challenge ("eustress") [9]

that can build resilience and behavioural flexibility. Figure 5 illustrates, using example data, time spent engaging with less valuable enrichment (a) and more beneficial enrichment (b), which promotes positive behaviour change for longer and describes a suitable example of an enrichment study timeframe (c) against a timeframe that is more likely to result in poor quality data due to confounding effects of time (d).

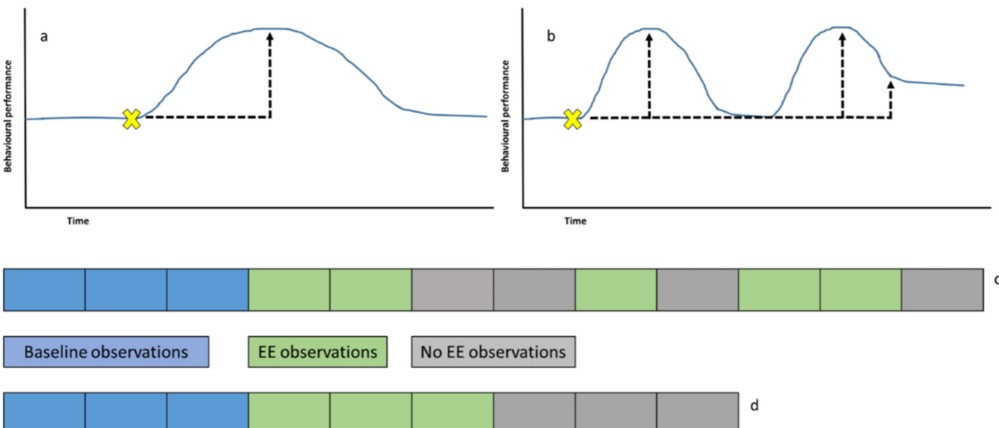

**Figure 5.** (**a**) Measuring the effect of one environmental enrichment (EE) device (yellow cross) that does not promote long-term beneficial change (as noted by the black dashed arrow) in activity. Once the animal has interacted with the EE and realised its potential or the EE has lost value, the behaviour returns to pre-EE baseline levels. (**b**) EE that has a prolonged or latent effect as the animal returns to a heightened state of arousal multiple times after the EE was originally provided. (**c**) Measurement of the latent effect of EE and evaluating long-term positive effects on behaviour is best conducted using a period of baseline (pre-EE) study compared to a random pattern of study days with and without EE to determine why behavioural performance alters over time when EE is present and when EE is removed (and if behavioural responses remain heightened once EE has gone). (**d**) Blocking observations into equal pre-EE, EE and then no-EE limits the conclusions that can be drawn on how long any effects of EE last as the animal may have lost interest in the EE during the green part of the data collection, and therefore, these data may be similar to that generated during the grey periods of study.

Enrichment studies are seminal to the development of evidence-based husbandry practice and are used to determine opportunities that the animal(s) finds useful/rewarding/important. We may anticipate an item to be enriching, but without the systematic study of behaviour change, we cannot determine if the item is actually enriching to the animal(s). As enrichment is added and removed over the duration of the study, the researcher must be mindful of extraneous variables (e.g., time of day, weather conditions, visitor presence and husbandry-related influences) and actively record them to maintain the validity of the research and its findings. Further reading on the use of enrichment for a wide range of zoo animals can be found in Young [87], and information on how to effectively evaluate the use of enrichment using the SPIDER principle (Set a goal, Planning, Implementing, Documenting, Evaluating and Readjusting) is available in Mellen and MacPhee [88]. When considering experimental work, Alligood et al. [89] expand on the range of behaviour-analytic methods suitable for designing a data collection procedure for an enrichment evaluation project.

### 3. Applying These Methods and Approaches to Practice: Using Observational Data to Audit Welfare

To provide an over-arching example of how these ten methods or theories can answer a specific question relating to zoo animal behaviour science, we evaluated their application to understanding and inferring animal welfare states. Collectively and individually, behavioural methodologies and theories can be used to examine animal welfare across

all zoo species. Behavioural observation allows inference of welfare states and supports consistent and standardised evaluation of welfare over time. Welfare is defined as the state of the individual as it attempts to cope with its environment [90] and encompasses both physical and psychological constructs [91]. Welfare is a continuum from positive through neutral to negative, and individuals will experience different welfare states throughout the course of the day [92]. The behaviours displayed by an animal are its responses to husbandry and the environment, influenced by mood and emotion. Experiences of previous environments, the animal's rearing history and its genetic characteristics will determine the responses of the specific individual animal and hence its current welfare state [8]. Measuring behaviour alongside assessment of the suitability, accessibility and ecological relevance of resources provides part of the information suitable for performing a welfare audit (i.e., scores of the quality of care provided based on evidence of husbandry and the animal's response to it) [75,93]. It is essential that appropriate behavioural measures are defined to ensure they are relevant to the species and situation under scrutiny, and more information on how to decide appropriate behavioural measures of zoo animal welfare is provided in Watters et al. [94]. Figure 6 examines how the methods and theories explained in Sections 2.1–2.10 can be applied to welfare auditing and the collection of behavioural evidence to inform welfare-positive husbandry.

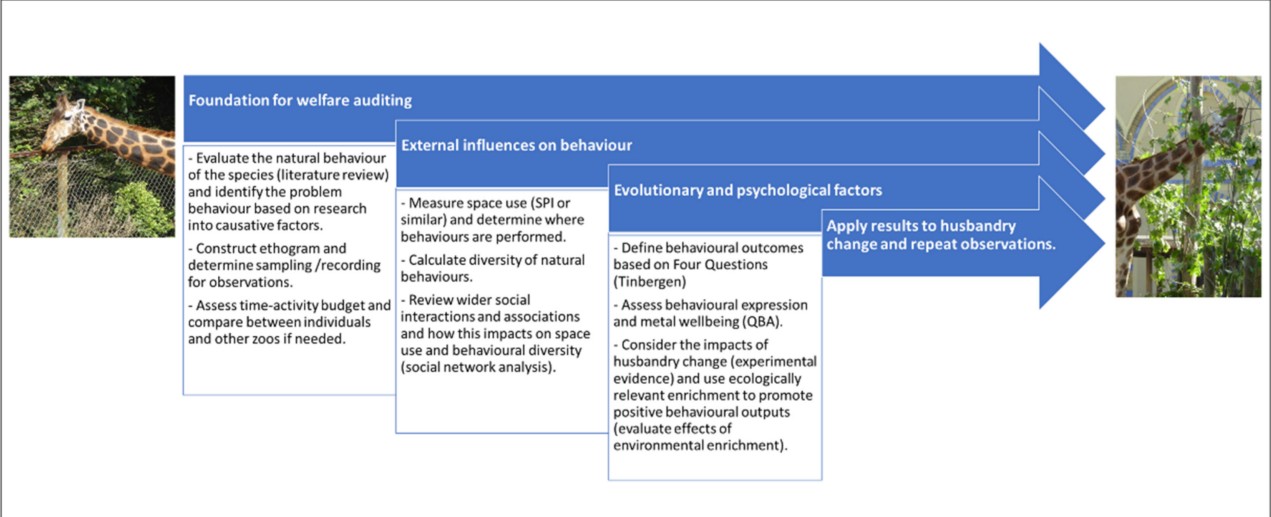

**Figure 6.** An example of including different methods, concepts and theories into an animal welfare research project. Commencing by supporting the aims and questions of the research with background research and a literature review to interpret behaviours, which builds into the framing and construction of the ethogram and behavioural recording/sampling techniques. Understanding the effects on welfare state from enclosure usage, the social environment and the diversity of behaviours performed by the individual and groups. Evaluating mental health and wellbeing by conducting a QBA to describe behavioural expression and reviewing the ecological relevance of all behaviours based on a review of Tinbergen's Four Question. Changing husbandry, e.g., by providing more outlets for a wider range of welfare positive behaviours to be performed and measuring the effect of this treatment compared to baseline conditions. Feedback of results directly into the zoo to affect husbandry and management and repeating the observational data collection period to determine the effect of this on the animals' welfare. Please note: not every project will require all of these methods to be included, and not all behavioural research is applied to husbandry change. This is simply an example of how a wide range of methods and theories could be used to better understand animal welfare as part of an applied research idea.

Meta-analyses and structured literature reviews help create and refine research questions, for example, identifying how behavioural studies could help improve animal welfare by deciphering why species perform abnormal repetitive behaviours [8]. The application of meta-analysis methodology to provide information on welfare is almost boundless. Identification of welfare positive behaviours (e.g., rumination in ruminant herbivores) that promote good psychological and physical health [95], gathering supporting evidence on

how much time is spent on this behaviour in a natural state (or similar) to interpret that seen in the zoo, and comparing across individuals to determine variation in responses to the zoo environment (and hence individual differences in coping) is a useful approach for the collection of behavioural data to infer welfare states. Ethogram construction, the definition of abnormal behaviours (e.g., artefacts of captivity and qualitative or quantitative difference from wild-type behaviours/time–activity budgets), quantification of how pervasive abnormal behaviour is in the time–activity budget of the individual or group and indices of behavioural diversity (how wide the behavioural repertoire is) are important first steps to understanding zoo animal welfare.

To promote good welfare via husbandry practice, one should know how space and resources are used, as well as which are preferred and valued by individual animals. This prevents human-biased assumptions from being made on what a zoo animal needs without evidence. Research on existing enclosure usage can be combined with research on how a change in environment causes a change in behaviour/space use to gain a fuller insight into provisions that may promote positive welfare states. As the social environment plays an important role in what behaviours are performed at the individual and group level [51] and hence impacts on the welfare state, social network analysis can provide helpful data for welfare assessment [57,60]. This is an especially useful tool when social groups are disrupted due to moves of individuals in or out of a group [96] or when differences in housing (indoors or outdoors) change the amount of space individuals have for performing preferential social behaviours [97]. Data from social network analysis can be used to identify areas of unwanted conflict or aggression within a group that may be over-exaggerated or unnatural in performance, and the group dynamic altered accordingly (e.g., by changing resource distribution) to enhance welfare for the individuals involved.

A challenging aspect of animal welfare measurement is deciphering the psychological aspects of welfare—explaining an animal's emotional states and its current feelings. QBA of affective states is especially useful to the furthering of animal welfare research [98] because it provides a way of examining emotional (internal) states. Emotional states are not directly observable and are therefore difficult to measure, but when using QBA, the style of behavioural expression (not what behaviours are performed but how those behaviours are performed) can be considered an indirect measurement of internal emotional state [71]. Welfare is a physical and psychological construct (and emotions are increasingly considered important to that construct); therefore, QBA is considered a progressive and useful behaviour-based tool for welfare assessment. Tinbergian reviews rely on the application of experiments to determine cause and effect relationships between the external environment (e.g., husbandry practices) and behaviour (welfare) change.

The application of experimental methods offers a vital opportunity to evidence construct validity—that a genuine welfare change is observed and that change is genuinely caused by the husbandry change. Counterbalanced repeated measures designs that measure behavioural change within a group of individuals are especially useful in this regard. Measurement of the latent effect of enrichment (i.e., how long positive behavioural change occurs once enrichment has been removed or has been used) allows assessment of how valuable the enrichment has been to promotion of positive welfare and should become more of a focus for EE-themed research. EE that keeps the animal interested over a longer timeframe is going to be of more value to the animal and more likely to promote positive welfare states by allowing the animal autonomy—giving control over its immediate environment and providing a choice of what it can do and when [99].

Without the control afforded in an experimental design, the cause of welfare change could not be isolated. Such control allows potential extraneous variables to be standardised across conditions or specifically measured so the effects of the extraneous variable can be isolated and distinguished from independent variable effects. As animals in the zoo are exposed to the presence of numerous people on a daily basis, consideration of any related impacts on behaviour should be measured when attempting to infer a welfare state [100]. The visitor effect is highly influenced by other variables (e.g., weather, temperature, pres-

ence of animal care staff, season and time of day [32,101,102]), and it is important for these variables to be recorded at the same time as the animal's behaviour to provide context to any potential influence of visitors. Comparative analysis of independent contrasts can also be used to help analyse species differences in behavioural responses [103] by controlling for non-independence caused species' evolutionary relatedness. A useful example assessing zoo animal behaviour and welfare whilst including data on phylogenetic relationships is provided by Mellor et al. [104].

Ultimately, whilst behaviour is a great tool for providing inferences on welfare states [105], the best practice is to combine such observational measures with other metrics (e.g., physiological sampling) to provide a more complete picture of the underlying welfare state [106,107]. Whilst this mixed-methods approach may be beyond the scope of smaller zoo- or animal-specific case studies, we include as a reminder of the limitations of behaviour-only welfare inferences and to provide "food for thought" for research extensions or longer-term studies, which could support behavioural data with other welfare measurements.

## 4. Discussion

This paper has outlined key behavioural methods, theories and concepts that can be used by zoo researchers to generate impactful, valid, and reliable data useful to advancing scientific output and developing zoo husbandry. We capture the enormous potential impacts of applying behavioural methodologies in zoo research. There are many other methodologies that we have not explored, such as preference and motivation testing and personality testing. Such methods tend to have specific applications but should never be unwarrantedly dismissed in the research planning stage. We hope that this paper informs students, researchers and zoo professionals about the array of observational methods available and the suitability of those methods for the research project being planned. Although zoo behavioural research is often formed of small N case studies [89,108], these still yield useful, applicable insights into zoo animal husbandry and management; telling researchers about natural history-based husbandry [109] and informing on population-specific responses to stressors in the zoo [110], for example. Behavioural data give an important insight into an individual's responses to the environment, allowing easily gained inferences on welfare state, mental and physical health, reproductive potential and growth and development.

As an expression of behaviour is unique to an individual and welfare is a multifaceted construct, combining methods offers the greatest opportunity to be holistic in the conclusion that can be drawn. Bringing together methods within a project to include calculation of time–activity budgets (what are animals doing and why might they be motivated to do this?) along with an assessment of space use (what areas of an enclosure are most important for the performance of specific activities?) and including social networks analyses (how does the social environment affect the use of the physical environment and the way in which an individual uses space?) [20,111] builds up a larger picture of welfare states, the suitability of husbandry and expands the impact of the finished project. It is essential to combine some approaches and methodologies to maintain overall effectiveness—such as integration of repeated measures experimental design into an enrichment study, which aims to show positive welfare change using an ethogram-informed activity budget plus the calculation of behavioural diversity indices. Simply put, the more methods that are applied, the more data that can be collected, the more holistic the interpretations of these data can be. This ultimately leads to a greater understanding of welfare states and a more meaningful (to the animal) change to husbandry practice.

Such mixed-methods approaches may be best achieved via collaboration. Building professional relationships and collaborations between zoos provides opportunities for longitudinal and impactful behavioural research to be completed [112–114], which is beneficial to all stakeholders (human and animal). Cross-institutional studies allow an analysis of the effects of each zoo's environment on behavioural responses and can enable researchers

to generate larger datasets for study, covering a larger overall sample population. Whilst many of the methods outlined here can be conducted by hand using pen and paper, video recording, photographic analysis and computer programmes to score behaviour patterns can also be used. Freeware (free to download and use computer software) such as Behavioral Observation Research Interactive Software (BORIS) allows for electronic cataloguing and recording of behaviour from imported video footage or live observations [115], and such freeware may be useful for certain research designs with specific subjects. Here academics often pioneer the development and use of such resources, and collaboration with zoo practitioners affords training and continuous professional development opportunities. This, in turn, creates better research as all users of the methodologies are informed. While zoo practitioners are essential to the achievement of quality research output, often conceptualising the initial research questions and aims of a study and practically implementing the research, academics are similarly important to constructing the study with validity in mind, often having more experience of designing multi-method projects and statistical knowledge for data analysis.

Reference points are extremely important when considering behavioural data—what is "normal", "typical" or "acceptable"? Wild data can help evaluate what we commonly see in the zoo; Rose [116] evaluates the information available on wild flamingo activity patterns to consider how behavioural diversity and beneficial activity (e.g., foraging) could be promoted in the zoo. Such an approach can be the start of an empirical investigation; the collection of information from the literature identifies what is known and what gaps in knowledge are apparent, and then the research question and associated methods can be developed to gather data to fill the gap. Of course, in a repeated measures experiment, the animal provides its own baseline reference point, yet it is still helpful to consult the literature and identify wider reference points for other individuals or populations in similar captive environments (or for wild-living individuals too). Often zoo professionals lack access to the scientific literature, and here, academics with institutional access to journal depositories or similar can help provide relevant papers.

As for all research, consideration of the amount of data needed and how long to collect data is important. A key element of any behavioural observation strategy is to avoid pseudoreplication. For social networks research, longer-term projects where data are collected at a few single points over the course of several weeks or months provide a more reliable picture of social ties than multiple observations over a few days. The longer-term study allows an animal to naturally change social partners, and therefore any consistent recording of two individuals together is likely a reflection of true social choice. A similar principle is important for time–activity budget data. Collection of instantaneous data at multiple short sample points (e.g., recording every 10 s) for a species where behavioural states change infrequently is likely to over-inflate the amount of data collected. Whereas a longer sampling interval (e.g., 2 or 3 min) is more likely to allow behaviour to change naturally or actually be the state that the animal wishes to be in.

How much data to collect is a question often posed by behavioural researchers, and the amount of time spent on data collection will depend on the species, its activity patterns and the question being asked. Figure 7 compares data for a captive group of African wild dogs collected for a short time period in 2018 to a longer time period in 2019. All data were collected during morning observation sessions. Differences in the percentage of observations per behaviour are very similar for three key states (feeding, active, social), and therefore, in this case, for this species, a short period of data collection is likely to provide a reliable and valid picture of how the animal spends its time. However, data on the interaction with enrichment and out of sight are not present for both time periods, and therefore behaviour performed more rarely, or that is stimulated by a specific event at a specific time (e.g., husbandry event or similar), may be missed during the shorter observer period. Likewise, interpreting the importance of the animals being out of sight may be easier for the longer period of observation, as more context to such a behavioural response can be provided by the evaluation of other activities, compared to the more

limited amount of data from the shorter observation schedule. If the animal group or species has an activity budget that stays fairly constant over the period, then the smaller datasets may correspond to larger ones more readily. If behaviour varies day-to-day, for example, shorter periods of observation may miss important patterns or details. A pilot study that provides information on the temporal change to activity patterns is an essential part of planning behavioural research to refine how much experimental data are needed.

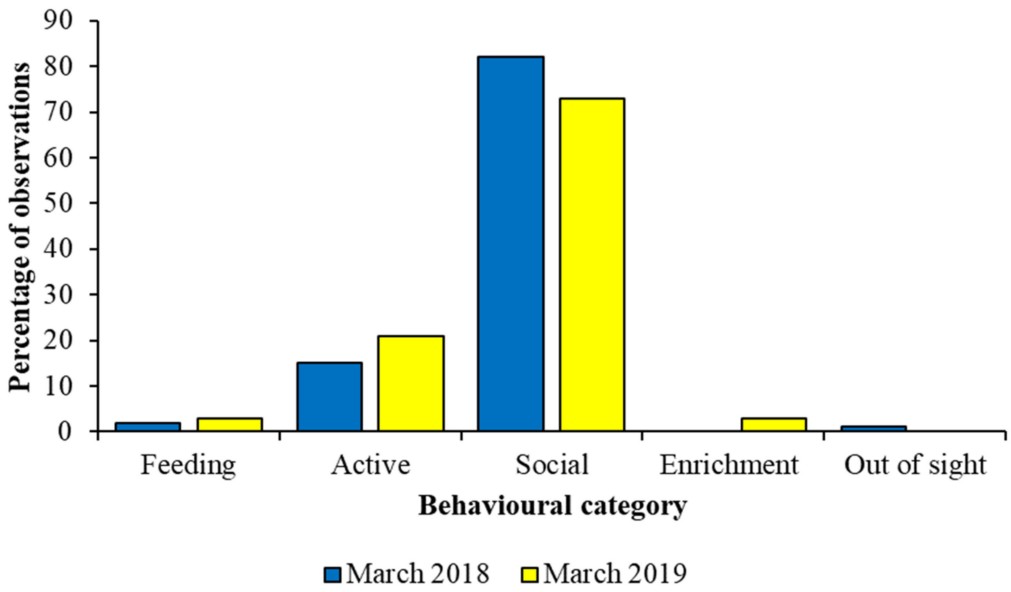

**Figure 7.** Behavioural data on captive African wild dog behaviours collected over seven days in 2018 provided a similar picture of activity to 26 days of data in 2019 for state behaviours that make up the majority of the species' time–activity budget. Data taken from McDonald et al. [117] and used with permission.

Data collected in the zoo may be considered unreliable or biased due to small sample sizes that typically limit generalisability. Consideration of the statistical analysis to choose the most appropriate test where the sample population is small [118], where the same animals have been the subject of repeated observations [108] confounding independence of data, and where the captive environment impacts what animals can and cannot do (and adds "noise" into a dataset) [119] is crucial for generating confidence in results. Animal husbandry is reliant on such data, and it is essential the researcher implements the most relevant behavioural observation and experimental design approaches to generate the fullest confidence in all data collected.

## 5. Conclusions

This paper has explained key methods, theories or concepts in animal behaviour that are available to the zoo scientist. These techniques are useful to developing ways of collecting valid and reliable observational data to answer a variety of questions relating to the responses of the animal(s) to the zoo. We have directed the researcher to appropriate sources of information that provide more details on these methods and explain each one in turn to enable the development of research methods and experimental design that give a strong foundation for good quality zoo science. Mixed-methods approaches, e.g., considering time–activity budgets, describing welfare states and social networks, can be used to maximise the output from behavioural study and provide a range of useful insights to the zoo on their animals. Behavioural research is an essential and useful component of the scientific outputs of zoos and aquariums, and we encourage researchers at all levels to continue to develop questions and ideas relating to animal behaviour to further the collection of husbandry evidence. To achieve the fullest understanding of behaviour and animal welfare from the application of observational methods and

theories, researchers need to: (1) combine methodologies; (2) work collaboratively; (3) be mindful to perform robust research that has been planned, implemented and analysed appropriately so that there is validity to interpretations and conclusions; (4) (if required from the research) feedback evidence that is appropriate to enacting relevant husbandry or management change.

**Author Contributions:** Conceptualization, P.E.R. and L.M.R.; writing—original draft preparation, P.E.R.; writing—review and editing, L.M.R. Both authors have read and agreed to the published version of the manuscript.

**Funding:** This research received no external funding.

**Institutional Review Board Statement:** Not applicable.

**Informed Consent Statement:** Not applicable.

**Data Availability Statement:** Not applicable.

**Conflicts of Interest:** The authors declare no conflict of interest.

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
