# Peer review of "Conducting Behavioural Research in the Zoo: A Guide to Ten Important Methods, Concepts and Theories"

_2673-5636, doi:10.3390/jzbg2030031_

Round 1

Reviewer 1 Report

The manuscript identifies and describes 10 behavioral research methodologies for use in a zoo/aquarium setting, providing practical applications for the different research designs. Generally, clear and concise wording, although a few longer sentences could be broken into 2+ to facilitate meaning.

Lines 37-45: References are needed for these sentences.

Lines 54-59: There are many published studies that support the benefits/uses of behavioural studies. Providing references would offer further resources for students reading this manuscript.

Line 92: There appears to be a word missing in the sentence—‘how long needs to spent’.

Line 128: Missing ‘of’ in between “definition eating”.

Lines 134-139: I agree that there is importance in compiling exhaustive behavioural ethograms depending on research questions/aims, but there is also value in ethograms that focus on higher order behaviors only.

Lines 162-165: The importance of inter-observer reliability should be expanded on, if only to give a clearer definition of this method. If this manuscript is geared towards students/early career individuals, many might not understand what this is and why it is so important in behavioural studies.

Lines 201-205: It is not directly clear what the authors mean here. Consider rewording or expanding on this thought to better tie in with previous sentences.

Line 202: A comma is needed after “[16]”

Line 206: Further emphasize is needed on the use of the same ethogram in longitudinal studies. As written, it can be interpreted that use of any ethogram will do.

Line 305: the researcher vs. the research?

Line 334: connect vs connected?

Line 344-345: review the clause starting after i.e.—it reads as if there is a word missing.

Line 377-378: Review the sentence for correct word choice. Consider removing ‘way’ and changing interaction to interact.

Line 478: comma is unnecessary.

Line 483: adding a comma in between “added was” would make the sentence clearer.

Lines 516-520: The sentence should be spilt into two to make it clearer, i.e. the example provided should be its own sentence.

Line 572-577: Sentence is difficult to understand—consider rewording/simplifying.

Line 679: missing closing quote.

Inferences of underlying welfare state/discussion: Watters, Krebs, Eschmann (2021) provides additional considerations associated with using behavioural research to assess welfare of zoo animals.

Table 1: There is a lot of text in the table, making it difficult to digest. Maybe make the symbols bold and further reading could just be a reference listed with the formula name. E.g., Spread of Participation Index [24, 26-27].

Author Response

Replies to reviewer 1

The manuscript identifies and describes 10 behavioral research methodologies for use in a zoo/aquarium setting, providing practical applications for the different research designs. Generally, clear and concise wording, although a few longer sentences could be broken into 2+ to facilitate meaning.

Thank you for the helpful and encouraging comments on the manuscript. We have reviewed the sentences within the document and tried to be more concise.

Lines 37-45: References are needed for these sentences.

We have included references to support these definitions.

Lines 54-59: There are many published studies that support the benefits/uses of behavioural studies. Providing references would offer further resources for students reading this manuscript.

We have included more references into this section.

Line 92: There appears to be a word missing in the sentence—‘how long needs to spent’.

Thank you. We have edited this.

Line 128: Missing ‘of’ in between “definition eating”.

We have edited this sentence to improve clarity.

Lines 134-139: I agree that there is importance in compiling exhaustive behavioural ethograms depending on research questions/aims, but there is also value in ethograms that focus on higher order behaviors only.

Thank you for the comment. We have edited this section to make it clearer that more case-study approaches (to an ethogram) are also relevant.

Lines 162-165: The importance of inter-observer reliability should be expanded on, if only to give a clearer definition of this method. If this manuscript is geared towards students/early career individuals, many might not understand what this is and why it is so important in behavioural studies.

Thank you. We have included more information on these methods here.

Lines 201-205: It is not directly clear what the authors mean here. Consider rewording or expanding on this thought to better tie in with previous sentences.

Thank you for the comments. We have edited this section for clarity to explain this application for time budgets.

Line 202: A comma is needed after “[16]”

Edited

Line 206: Further emphasize is needed on the use of the same ethogram in longitudinal studies. As written, it can be interpreted that use of any ethogram will do.

Thank you for spotting this. We have edited accordingly.

Line 305: the researcher vs. the research?

Edited.

Line 334: connect vs connected?

Edited to “are connected…”

Line 344-345: review the clause starting after i.e.—it reads as if there is a word missing.

Thank you for spotting this. We have edited this sentence to split it so that the clause is now correct.

Line 377-378: Review the sentence for correct word choice. Consider removing ‘way’ and changing interaction to interact.

Thank you for spotting this. We have edited for clarity.

Line 478: comma is unnecessary.

We have edited this sentence for clarity.

Line 483: adding a comma in between “added was” would make the sentence clearer.

Edited.

Lines 516-520: The sentence should be spilt into two to make it clearer, i.e. the example provided should be its own sentence.

Edited accordingly.

Line 572-577: Sentence is difficult to understand—consider rewording/simplifying.

Edited this section for clarity.

Line 679: missing closing quote.

Thank you for spotting this. We have edited.

Inferences of underlying welfare state/discussion: Watters, Krebs, Eschmann (2021) provides additional considerations associated with using behavioural research to assess welfare of zoo animals.

Thank you. we have included this reference within the text in the inferences of welfare state section.

Table 1: There is a lot of text in the table, making it difficult to digest. Maybe make the symbols bold and further reading could just be a reference listed with the formula name. E.g., Spread of Participation Index [24, 26-27].

Thank you for the suggestion. We have edited accordingly.

Reviewer 2 Report

The purpose of this paper is to provide students and those who are new to behavioral research an overview of the methodologies used in behavioral research, including examples of the types of questions that can be addressed using specific methods. I think this paper has the potential to be a valuable resource, but major revisions are needed. Overall, I’m concerned about the organization and readability of the paper. I understand the length of these comments may be daunting, please take it as a sign that I think this is an important work that should move forward and want to be of as much assistance as possible as a reviewer.

First, I think the paper needs considerable restructuring. I appreciate that the authors emphasize multiple methods can be used together. However, I disagree that all ten items listed are actual methodological approaches, or perhaps the title is misleading. Based on the title, I expected a review of things like behavioral diversity and space use; I disagree that evaluating effects on enrichment, experimental studies and repeated measures, or a Tinbergian approach are actual behavioral methods. If I want to know the effects of enrichment, I put together an ethogram, use an experimental study with repeated measures, and investigate changes in activity budget and/or space/resource use. The authors conflate study design (e.g., important considerations for making an ethogram, having pre, during, and post observations, etc.) with the methods (activity budget, QBA, space use). It would require a change in title and major restructuring of the paper but I think there is definitely value in an introductory source that talks about methods and design considerations together, I just don’t think they should all be lumped in the same section. As I mention later on in the review, I think in the revised section with methods, examples of potential research topics as well as how those methods can provide insight into welfare should be integrated, rather than welfare inferences as a standalone section.

Given that this paper is primarily aimed at students and those who are new to behavioral research, I would recommend substantial editing for readability. I am familiar with several of the first author’s previously published papers and have always found their work to be highly readable and easy to understand, but the text in this manuscript tends to be dense and has frequent inappropriate comma usage that inhibits easy comprehension (the first two sentences of the paper highlight this but it is found throughout). Many of the systems are also very long. I have pointed it out in some cases but I would recommend the authors reviewing the entire manuscript for places where they can simplify the writing.

Throughout the section on behavioral methods, the authors are inconsistent in mentioning example studies that use the methods being discussed (for example, several are listed for EIs but not SPIs in section 2.3). Rather than sporadically mentioning example studies, I recommend the authors put together a table with each behavioral method and a few examples of papers which use those methods (this doesn’t need to be extensive, I think 3-5 examples for each method would be sufficient, as interested readers can then look to those for more information). If the authors take my suggestion for restructuring the content, the same could be done for study design (e.g., 3-5 papers showing good repeated measures study design in zoo research).

Introduction

Lines 37 - The start of this sentence is difficult to read. I would recommend just “individuals” and removing the “human or non-human animals”. If you want to keep that phrase, I would suggest removing the commas and making humans plural, so “...of individual humans or non-human animals to endogenous…”

Line 40 - Remove comma after “alike”

Line 46 - Remove comma after “manipulated”

Line 49 - Remove comma after “data”

Lines 59-61 - I’m not sure “behaviour-based” works at the beginning of this sentence - of course behavior is a consistent focus and method in behavior-based zoo research. I think you can remove that phrase from the start to instead state that zoo research has expanded in remit but behavior remains a focus and method.

Line 88 - Remove comma after “cameras”

Line 93 - Recommend “open” rather than “opening”

Behavioral Methods

Line 129 - Remove comma after “observe”

Line 137 - Nishida, Zamma… and colleagues? Et al? This happens frequently in the paper, unless it’s how due to journal formatting I would edit them throughout. 

Line 145 - Remove comma after “budgets”

Line 149 - I recommend removing the comma after “days” and inserting the word “and” instead.

Line 150 - Remove comma after “defined”

Line 160 - Add comma after “sources”

Line 162 - Remove comma after “field”

Overall comment on section 2.1 - I am concerned that this section on ethograms is incredibly daunting to new behavioral researchers. I would encourage the authors to better emphasize, from early on in this section, that ethograms can be brief and tailored to questions of interest. By mentioning the Nishida et al. work on chimpanzees as an example, ethograms sound like an impossible task. I think it would be reassuring to researchers to discuss how some ethograms are very detailed with many behaviors and others are far more generalized and may only have a few (including an “other” category for rare behaviors or a “none” category for recording when the animal is doing a behavior that isn’t relevant to the current study), and that it is very much based on the question. It is important to emphasize that careful work needs to go into designing an ethogram but not all studies that use an ethogram require documenting every single minute behavior. Perhaps better differentiate between a comprehensive behavioral inventory (such as the Nishida et al. work) and the focused ethogram that someone would use in a study. I also believe including an ethogram or two from the authors’ own work would be beneficial as an example, rather than just referencing one.

Line 202 - Comma after “[16]”

Line 229 - Typo - “combing” should be “combining”; remove comma after “measurement”

Lines 250-252 - Inconsistent formatting for species names

Lines 256-261 - I’d recommend merging these two paragraphs.

Line 296 - Remove comma after “observed”

Lines 299-300 - Do you mean research is necessary to determine biologically relevant behaviors? Also the researchers would make the ethogram that they then use in their study to calculate biological diversity, so perhaps this sentence should reference research into the natural behaviors of the species or previous zoo research on that species prior to data collection to determine essential behaviors for the ethogram? To me that seems logical given the next sentence.

Line 307 - I recommend using the word “should” instead of “must”

Line 308 - I would recommend ending this with “and provide data on animal welfare states” or something similar. There is still a large debate around abnormal behavior and what it means for welfare, but the way the sentence reads currently implies that abnormal behavior signals poor welfare. There are cases where abnormal behavior has likely been socially learned and thus does not signal poor welfare, or cases where abnormal behavior benefits the animal by helping it cope. This article is certainly not the place to go into those nuances but I would recommend editing the end of the sentence to not give the wrong impression to new researchers who may not be familiar with this area.

Line 309 - Formula should be plural

Lines 312-314 - Please expand on this. I also think it would fit better as the final sentence (with another 1-2 sentences expanding) at the end of the paragraph in lines 321-327, rather than attached here with the sentence on formulas (this sentence on formulas could easily be the ending sentence of the previous paragraph, so then it wouldn’t be standalone).

Line 324 - Animal should be plural

Line 325 - I recommend removing “and” and making the section starting “modification of…” it’s own sentence.

Line 334 - Remove “that”; are the connecting lines technically called edges? Please check this. I’m not an expert in social network analysis but considering the lines can cross one another and are often presented like a web, it seems strange they would be called edges.

Line 346 - I would recommend not starting this sentence with “And”, it doesn’t seem necessary.

Line 369 - Remove “too” from the end of the sentence.

Line 378 - “way” should be “which”; “interactions” should be “interacts”

Line 387 - This sentence is difficult to follow, I recommend putting the name of the technique and citation in parentheses.

Line 390 - “observe” rather than “observer”

Lines 392-393 - Reliability scoring when there are multiple observers is important for any behavioral method, so I’m not sure mentioning it here specifically for QBA is needed. It should instead be a more general consideration to be mentioned in the introduction or discussion. Or, if restructuring, perhaps in the section on study design considerations.

Line 483 - Add a comma after “added”

Line 485 - Is EV a standard abbreviation? I haven’t seen it before and so immediately forgot it after I saw it, then became confused when I read it again later on (my first thought was endogenous or exogenous variable). There are a lot of abbreviations in this paper already, I’d recommend removing this one and just using “extraneous variable” instead.

Lines 504-507 - Thank you for providing a simple explanation of fixed vs. random effects, I find these are confusing to many!

Line 515 - Remove comma after “positive”

Line 518 - The “but continuous” does not seem to fit here, I would recommend removing it since the meaning of the sentence doesn’t change without it.

Inferences of underlying welfare state

Overall, this section does not seem to flow. The authors already mention some research ideas in the previous section when describing each methodological approach. I think the paper would flow better if ways that these methods might provide insight into welfare were incorporated into those areas rather than in this standalone section.

Line 573 - Insert a comma after “herbivores”; promotes should be singular

Line 601 - Remove comma after “group”

Line 616-619 - Given that a QBA meta-analysis isn’t at all possible and won’t be for a long time, I’m not sure this is a relevant statement that really belongs in an introductory paper. If the authors choose to keep it, “required” in line 618 should be “requires”.

Discussion

Line 670 - Remove comma after “studies”

Line 679 - Is there supposed to be a quotation mark around “good”?

Line 694 - Insert comma after the second “data”

Lines 705-707 - Again I don’t think emphasizing interobserver reliability for QBA - when it is not emphasized for behavioral research with multiple observers overall - is appropriate. For those not familiar with behavior research, the way the manuscript currently reads is that interobserver reliability is important for QBA but not anything else because it isn’t mentioned elsewhere. While I could see it appropriate to especially emphasize in the QBA section that interobserver reliability can function as a way of making “subjective” descriptors and scores more “objective” (so long as interobserver reliability is discussed more broadly as well), I don’t think any of that is needed here.

Lines 711-713 - I’m not sure what this sentence means.

Lines 740-751 - This paragraph makes some very important points!

Line 741 - Remove the word “for”

Line 744 - Change “that” to “than”

Lines 752-764 - Is there any guidance in the literature at all on the number of hours of behavioral data to collect? Like “each phase of a project needs at least 20 hours per focal animal” or something of that sort? It is something I’ve always wondered myself but have never seen an answer so I’m curious! 

Line 773 - Remove the word “on”

Conclusions

Line 785 - “welfare inferences” is not a method

Author Response

Replies to reviewer 2

The purpose of this paper is to provide students and those who are new to behavioral research an overview of the methodologies used in behavioral research, including examples of the types of questions that can be addressed using specific methods. I think this paper has the potential to be a valuable resource, but major revisions are needed. Overall, I’m concerned about the organization and readability of the paper. I understand the length of these comments may be daunting, please take it as a sign that I think this is an important work that should move forward and want to be of as much assistance as possible as a reviewer.

Thank you for the detailed and useful comments. We have edited and reviewed each section of the paper to ensure that it is now clearer to read and easier to understand.

First, I think the paper needs considerable restructuring. I appreciate that the authors emphasize multiple methods can be used together. However, I disagree that all ten items listed are actual methodological approaches, or perhaps the title is misleading. Based on the title, I expected a review of things like behavioral diversity and space use; I disagree that evaluating effects on enrichment, experimental studies and repeated measures, or a Tinbergian approach are actual behavioral methods. If I want to know the effects of enrichment, I put together an ethogram, use an experimental study with repeated measures, and investigate changes in activity budget and/or space/resource use. The authors conflate study design (e.g., important considerations for making an ethogram, having pre, during, and post observations, etc.) with the methods (activity budget, QBA, space use). It would require a change in title and major restructuring of the paper but I think there is definitely value in an introductory source that talks about methods and design considerations together, I just don’t think they should all be lumped in the same section. As I mention later on in the review, I think in the revised section with methods, examples of potential research topics as well as how those methods can provide insight into welfare should be integrated, rather than welfare inferences as a standalone section.

Thank you for the comments. We have changed the title to better describe the content of the paper. We are aiming to provide an explanation of tools that can be used to create a suitable experimental design for behavioural research in the zoo. We have included methods (such as how to collect behavioural data, how to measure enclosure usage and the like), concepts (such as how and why we should evaluate environmental enrichment) and approaches (for example a Tinbergian approach to understanding animal behaviour or how to infer welfare state from behavioural data).

We have edited each section heading in section 2.X to make it clearer what that chapter contains. We have also included information from section 3 to provide more information on the methods and approaches explained here.

Given that this paper is primarily aimed at students and those who are new to behavioral research, I would recommend substantial editing for readability. I am familiar with several of the first author’s previously published papers and have always found their work to be highly readable and easy to understand, but the text in this manuscript tends to be dense and has frequent inappropriate comma usage that inhibits easy comprehension (the first two sentences of the paper highlight this but it is found throughout). Many of the systems are also very long. I have pointed it out in some cases but I would recommend the authors reviewing the entire manuscript for places where they can simplify the writing.

Thank you for the helpful suggestions to edit sentence structure. We have reviewed the whole manuscript and checked the sentence structure throughout. We apologise for the long sentence and overuse of commas, we were attempting to provide examples and definitions as we went but we can see that this approach has caused confusion and a lack of clarity.  

Throughout the section on behavioral methods, the authors are inconsistent in mentioning example studies that use the methods being discussed (for example, several are listed for EIs but not SPIs in section 2.3). Rather than sporadically mentioning example studies, I recommend the authors put together a table with each behavioral method and a few examples of papers which use those methods (this doesn’t need to be extensive, I think 3-5 examples for each method would be sufficient, as interested readers can then look to those for more information). If the authors take my suggestion for restructuring the content, the same could be done for study design (e.g., 3-5 papers showing good repeated measures study design in zoo research).

Thank you for the comment. We are a bit confused by this comment as in Table 1, for section 2.3, where we explain each enclosure use method we included example studies for the two SPI  formula and the EI formula. We have made sure that each section contains examples of the method or concept in practice and we provide full references for the reader to follow up if they wish. We moved the references with Table 1 to be included with the name of the method, based on the suggestion of another reviewer (to make this table clearer) so we have provided an outline (“introduction”) to section 2 to explain to the reader what information is provided in this section.  We have also taken on board your suggestion to include information from section 3 into section 2 and hopefully this provides more examples of methods or theories in practice.

Introduction

Lines 37 - The start of this sentence is difficult to read. I would recommend just “individuals” and removing the “human or non-human animals”. If you want to keep that phrase, I would suggest removing the commas and making humans plural, so “...of individual humans or non-human animals to endogenous…”

We have edited this comment accordingly.

Line 40 - Remove comma after “alike”

 Edited

Line 46 - Remove comma after “manipulated”

 Edited

Line 49 - Remove comma after “data”

 Edited

Lines 59-61 - I’m not sure “behaviour-based” works at the beginning of this sentence - of course behavior is a consistent focus and method in behavior-based zoo research. I think you can remove that phrase from the start to instead state that zoo research has expanded in remit but behavior remains a focus and method.

Thank you for the suggestion. We have edited accordingly. 

Line 88 - Remove comma after “cameras”

Edited  

Line 93 - Recommend “open” rather than “opening”

Thank you for the suggestion. We have changed to “zoo’s hours of opening”.

Behavioral Methods

Line 129 - Remove comma after “observe”

Edited

 Line 137 - Nishida, Zamma… and colleagues? Et al? This happens frequently in the paper, unless it’s how due to journal formatting I would edit them throughout. 

We have tried to edit the format to include the first author’s surname and et al. for the citations. But we are now not sure if contravenes the journal’s style.

Line 145 - Remove comma after “budgets”

 Edited

Line 149 - I recommend removing the comma after “days” and inserting the word “and” instead.

This section has since been re-written and this is now removed.

Line 150 - Remove comma after “defined”

Edited. 

Line 160 - Add comma after “sources”

This section has since been re-written and this is now removed.

Line 162 - Remove comma after “field”

 Edited

Overall comment on section 2.1 - I am concerned that this section on ethograms is incredibly daunting to new behavioral researchers. I would encourage the authors to better emphasize, from early on in this section, that ethograms can be brief and tailored to questions of interest. By mentioning the Nishida et al. work on chimpanzees as an example, ethograms sound like an impossible task. I think it would be reassuring to researchers to discuss how some ethograms are very detailed with many behaviors and others are far more generalized and may only have a few (including an “other” category for rare behaviors or a “none” category for recording when the animal is doing a behavior that isn’t relevant to the current study), and that it is very much based on the question. It is important to emphasize that careful work needs to go into designing an ethogram but not all studies that use an ethogram require documenting every single minute behavior. Perhaps better differentiate between a comprehensive behavioral inventory (such as the Nishida et al. work) and the focused ethogram that someone would use in a study. I also believe including an ethogram or two from the authors’ own work would be beneficial as an example, rather than just referencing one.

Thank you for the comment. We agree that this reads as too complicated. We have simplified the explanation of what an ethogram is, provided an example as suggested and moved the Nishida et al. example of a complex and comprehensive ethogram to the end of the section. If you feel that this example is still unhelpful and complicated, we can remove it.  

Line 202 - Comma after “[16]”

Edited  

Line 229 - Typo - “combing” should be “combining”; remove comma after “measurement”

Thank you for spotting this, we have edited. 

Lines 250-252 - Inconsistent formatting for species names

Thank you for the comment but we are unsure of the issue here?

Lines 256-261 - I’d recommend merging these two paragraphs.

 Edited.

Line 296 - Remove comma after “observed”

Edited. 

Lines 299-300 - Do you mean research is necessary to determine biologically relevant behaviors? Also the researchers would make the ethogram that they then use in their study to calculate biological diversity, so perhaps this sentence should reference research into the natural behaviors of the species or previous zoo research on that species prior to data collection to determine essential behaviors for the ethogram? To me that seems logical given the next sentence.

Thank you for comment, we have edited this section to include the information suggested above.

Line 307 - I recommend using the word “should” instead of “must”

 Edited. Thank you for the suggestion.

Line 308 - I would recommend ending this with “and provide data on animal welfare states” or something similar. There is still a large debate around abnormal behavior and what it means for welfare, but the way the sentence reads currently implies that abnormal behavior signals poor welfare. There are cases where abnormal behavior has likely been socially learned and thus does not signal poor welfare, or cases where abnormal behavior benefits the animal by helping it cope. This article is certainly not the place to go into those nuances but I would recommend editing the end of the sentence to not give the wrong impression to new researchers who may not be familiar with this area.

Thank you for the suggestion. We have edited to state that such measures can be indicative of welfare states.

Line 309 - Formula should be plural

 Edited

Lines 312-314 - Please expand on this. I also think it would fit better as the final sentence (with another 1-2 sentences expanding) at the end of the paragraph in lines 321-327, rather than attached here with the sentence on formulas (this sentence on formulas could easily be the ending sentence of the previous paragraph, so then it wouldn’t be standalone).

Thank you for the suggestion. We have moved this sentence to the end of the paragraph as suggested as changed the use of the reference and included more detail.

Line 324 - Animal should be plural

Edited

Line 325 - I recommend removing “and” and making the section starting “modification of…” it’s own sentence.

 Edited

Line 334 - Remove “that”; are the connecting lines technically called edges? Please check this. I’m not an expert in social network analysis but considering the lines can cross one another and are often presented like a web, it seems strange they would be called edges.

 Edge is the correct term for the connection between nodes.

Line 346 - I would recommend not starting this sentence with “And”, it doesn’t seem necessary.

This section is now altered based on other reviewer comment and the issue has been resolved.

Line 369 - Remove “too” from the end of the sentence.

 As above.

Line 378 - “way” should be “which”; “interactions” should be “interacts”

 As above.

Line 387 - This sentence is difficult to follow, I recommend putting the name of the technique and citation in parentheses.

Thank you for the suggestion, we have changed accordingly. 

Line 390 - “observe” rather than “observer”

 Edited

Lines 392-393 - Reliability scoring when there are multiple observers is important for any behavioral method, so I’m not sure mentioning it here specifically for QBA is needed. It should instead be a more general consideration to be mentioned in the introduction or discussion. Or, if restructuring, perhaps in the section on study design considerations.

Thank you for the comment. IOR is now clarified and explained in more detail earlier in the manuscript (to emphasis its importance to general observational data collection). We have included this here (and wish to retain) because it is crucial to how QBA works (in terms of agreement with definitions and descriptions) for IOR to be part of the development of term lists used for welfare assessment.  

Line 483 - Add a comma after “added”

 Edited.

Line 485 - Is EV a standard abbreviation? I haven’t seen it before and so immediately forgot it after I saw it, then became confused when I read it again later on (my first thought was endogenous or exogenous variable). There are a lot of abbreviations in this paper already, I’d recommend removing this one and just using “extraneous variable” instead.

Thank you for the suggestion. Edited accordingly. We have also removed IV and DV to be clearer.

Lines 504-507 - Thank you for providing a simple explanation of fixed vs. random effects, I find these are confusing to many!

We are glad that you found this useful. 

Line 515 - Remove comma after “positive”

Edited. 

Line 518 - The “but continuous” does not seem to fit here, I would recommend removing it since the meaning of the sentence doesn’t change without it.

Edited  

Inferences of underlying welfare state

Overall, this section does not seem to flow. The authors already mention some research ideas in the previous section when describing each methodological approach. I think the paper would flow better if ways that these methods might provide insight into welfare were incorporated into those areas rather than in this standalone section.

Thank you for the comment. in line with other reviewers we have reduced this section, to provide an overview of using all of the 10 methods or theories to answering a specific question (i.e. measuring welfare) and we have integrated the remaining information in section 2.X. We have also included a new diagram (now Figure 6) that explains how the methods and concepts examined can all be used in practice to answer an example question. If you are still unhappy with this example of applying the methods/approaches to welfare measurement, we are happy to remove.

Line 573 - Insert a comma after “herbivores”; promotes should be singular

Edited.  

Line 601 - Remove comma after “group”

Edited  

Line 616-619 - Given that a QBA meta-analysis isn’t at all possible and won’t be for a long time, I’m not sure this is a relevant statement that really belongs in an introductory paper. If the authors choose to keep it, “required” in line 618 should be “requires”.

Thank you for the suggestion. We have deleted this section. 

Discussion

Line 670 - Remove comma after “studies”

Edited  

Line 679 - Is there supposed to be a quotation mark around “good”?

 We have removed these and altered the sentence.

Line 694 - Insert comma after the second “data”

Edited.  

Lines 705-707 - Again I don’t think emphasizing interobserver reliability for QBA - when it is not emphasized for behavioral research with multiple observers overall - is appropriate. For those not familiar with behavior research, the way the manuscript currently reads is that interobserver reliability is important for QBA but not anything else because it isn’t mentioned elsewhere. While I could see it appropriate to especially emphasize in the QBA section that interobserver reliability can function as a way of making “subjective” descriptors and scores more “objective” (so long as interobserver reliability is discussed more broadly as well), I don’t think any of that is needed here.

Thank you for the comment. We have deleted this section.

Lines 711-713 - I’m not sure what this sentence means.

We have deleted this section. 

Lines 740-751 - This paragraph makes some very important points!

Thank you for the positive comments.

Line 741 - Remove the word “for”

 Edited

Line 744 - Change “that” to “than”

Edited  

Lines 752-764 - Is there any guidance in the literature at all on the number of hours of behavioral data to collect? Like “each phase of a project needs at least 20 hours per focal animal” or something of that sort? It is something I’ve always wondered myself but have never seen an answer so I’m curious! 

We searched and searched for this, to but to no avail. Hence why we decided to include the information on the hunting dog project to show an example of different timeframes.  

Line 773 - Remove the word “on”

 Edited

Conclusions

 Line 785 - “welfare inferences” is not a method

Edited to be clearer that we have explained methods, concepts and theories.  

Reviewer 3 Report

COMMENTS

The manuscript is a good introduction to the study methods that can be carried out in a zoo. As the authors wrote, it is not an article that "explores all aspects" of potential research and its methods; it is an introduction that tries to address many of these aspects, with examples, potentialities, methods of analysis, relevant questions and so on. A good topic deserves to be published. However, I make some general suggestions before taking this publication further.

I think the manuscript is not a synopsis, and could not be, with a theme as vast as "Behavioural research in the zoo". There are 24 pages and 101 references!

I think the article should have more focus adhering to the authors' goal. From line 552 (Inferences of underlying welfare state) to line 659, it deals with a relevant topic (Animal Welfare), but without the focus of what the authors aim: "The aims of this paper are to provide a synopsis of important methodological approaches to the study of animal behavior in the zoo, to guide the end stages of methodological planning and the collection of valid and relevant data". I suggest that this other topic (Inferences of underlying welfare state) be addressed in another article, with the depth and experience with which the authors seem to have. However, I think there is a crucial text that should be used in this manuscript. That is, the approach to Tinbergen's four questions. From line 636 to line 659, there is a fundamental discussion for the study of animal behavior. This topic was discussed more superficially in the item "2.8. Tinbergen’s Four Questions" (Line 432). I suggest that the text from lines 636 to 659 be adapted and inserted in item 2.8.

Many references address similar topics. There are references redundantly cited. I quote some examples: Branch, 2019 and Baker, 2016; Bateson & Martin, 2021 and Lehner, 1987, and Kaufman & Rosenthal, 2009; Shepherdson et al., 1993 and Clark & Melfi, 2012; de Vere, 2018 and Harley, 2019; Goh et al., 2017 and Checon et al., 2020; References from numbers 44 to 50.

Another problem with the manuscript is the fluidity and comprehension of the text. In parts of the text, there is an excess of parentheses, with interruptions and fragmentations, hindering the pleasure of reading a complex text like this one, even when it is intended for "primarily students and new-to-the-zoo researchers”. Many long sentences make it difficult to understand.

Line 40: What do the authors mean by “subconsciously”?

Line 557: “Five Freedoms or Five Domains”. Please, inserts a reference.

Line 667-669: this sentence is a conclusion, not a discussion.

Line 765: Figure 5 does not seem to inform very well what is described in the text. It does not appear that the differences in the percentage of observations are similar. In fact, “enrichment” and “out of sight” are very different between periods. The text is well described about the timing of behavioral observations.

Author Response

Replies to reviewer 3

The manuscript is a good introduction to the study methods that can be carried out in a zoo. As the authors wrote, it is not an article that "explores all aspects" of potential research and its methods; it is an introduction that tries to address many of these aspects, with examples, potentialities, methods of analysis, relevant questions and so on. A good topic deserves to be published. However, I make some general suggestions before taking this publication further.

Thank you for the kind words and support of the paper, and the suggested edits, which we have actioned below.

I think the manuscript is not a synopsis, and could not be, with a theme as vast as "Behavioural research in the zoo". There are 24 pages and 101 references!

Thank you for the important comment. We have changed the title name. We struggled to think of a suitable way of describing the paper, so we have edited to “an explanation of…”

I think the article should have more focus adhering to the authors' goal. From line 552 (Inferences of underlying welfare state) to line 659, it deals with a relevant topic (Animal Welfare), but without the focus of what the authors aim: "The aims of this paper are to provide a synopsis of important methodological approaches to the study of animal behavior in the zoo, to guide the end stages of methodological planning and the collection of valid and relevant data". I suggest that this other topic (Inferences of underlying welfare state) be addressed in another article, with the depth and experience with which the authors seem to have. However, I think there is a crucial text that should be used in this manuscript. That is, the approach to Tinbergen's four questions. From line 636 to line 659, there is a fundamental discussion for the study of animal behavior. This topic was discussed more superficially in the item "2.8. Tinbergen’s Four Questions" (Line 432). I suggest that the text from lines 636 to 659 be adapted and inserted in item 2.8.

Thank you for comments. We have reduced the animal welfare section and included the information on Tinbergen in the main article. If you are still unhappy with the animal welfare example section we are happy to remove.

Many references address similar topics. There are references redundantly cited. I quote some examples: Branch, 2019 and Baker, 2016; Bateson & Martin, 2021 and Lehner, 1987, and Kaufman & Rosenthal, 2009; Shepherdson et al., 1993 and Clark & Melfi, 2012; de Vere, 2018 and Harley, 2019; Goh et al., 2017 and Checon et al., 2020; References from numbers 44 to 50.

Thank you for the comment. We have included a range of references that show the explanation of the theory or concept and then the application of method or concept to data collection in the zoo. We would like to keep in the range of references to give the reader the opportunity to read more widely around the subject.

Another problem with the manuscript is the fluidity and comprehension of the text. In parts of the text, there is an excess of parentheses, with interruptions and fragmentations, hindering the pleasure of reading a complex text like this one, even when it is intended for "primarily students and new-to-the-zoo researchers”. Many long sentences make it difficult to understand.

Thank you for the comments. We have reviewed the entire manuscript to remove ambiguous or unclear sentence structures.

Line 40: What do the authors mean by “subconsciously”?

We have edited to instinctively.

Line 557: “Five Freedoms or Five Domains”. Please, inserts a reference.

This example has since been deleted.

Line 667-669: this sentence is a conclusion, not a discussion.

Thank you for the comment. We have moved to the conclusion.

Line 765: Figure 5 does not seem to inform very well what is described in the text. It does not appear that the differences in the percentage of observations are similar. In fact, “enrichment” and “out of sight” are very different between periods. The text is well described about the timing of behavioral observations.

Thank you for the comment. We have edited to explain that for some short-term behaviours, short timeframes may be inappropriate

Round 2

Reviewer 2 Report

Thank you to the authors for responding to the feedback provided, I know it was lengthy and I appreciate their thorough consideration and responses to all of the comments. I have no major concerns and have provided only a handful of minor suggestions.

Line 78: “analyse” should be “analysis”

Lines 89-93: Since reviewing temporal or seasonal activity patterns is useful for deciding when to do either remote monitoring or live observation, I’d suggest a slight rewording here: “... will help with deciding when to place trail cameras (or similar technology for remote monitoring) or observe in person to provide the best chance of capturing maximal behavioral data.”

Lines 179-189: Thank you for adding this information!

Line 263: Remove comma after “measurement”

Line 292: Remove word “it” or add the word “is” after

Lines 343-344: This is the sentence where I suggested using something like “and provide data on animal welfare states” to end the statement rather than “can impact on animal welfare states” due to the debate around abnormal behavior and what it means for welfare.

Line 371: “then” should be “them”

Line 537: “designs” should be singular

Lines 606-607: Perhaps give an example or two of an extraneous variable in parentheses for those unfamiliar with the term.

Figure 6: I really like this and think it’s helpful. With introductory resources, however, I worry about some readers assuming they must do every single thing as written. Therefore, I would suggest removing reference to a meta-analysis from the figure itself as well as the caption. Done properly on species with a large amount of literature, meta-analyses are incredibly time consuming and imply the use of statistics which aren’t necessary if the researcher simply wants to review existing research and put together an ethogram of known behaviors. Meta-analyses are certainly powerful and should be used more often but they’re not necessary to begin a project. I’d simply suggest a “literature review” instead.

Line 752: As mixed-methods approaches can be (and frequently are) done at a single zoo, I’d recommend replacing “are” with “may be”

Author Response

Replies to reviewer 2

Thank you to the authors for responding to the feedback provided, I know it was lengthy and I appreciate their thorough consideration and responses to all of the comments. I have no major concerns and have provided only a handful of minor suggestions.

Thank you for the feedback and the extra suggestions for improvement, which we have actioned in the text.

Line 78: “analyse” should be “analysis”

Edited

Lines 89-93: Since reviewing temporal or seasonal activity patterns is useful for deciding when to do either remote monitoring or live observation, I’d suggest a slight rewording here: “... will help with deciding when to place trail cameras (or similar technology for remote monitoring) or observe in person to provide the best chance of capturing maximal behavioral data.”

Thank you for the suggestion, we have incorporated this accordingly.

Lines 179-189: Thank you for adding this information!

Thank you for the positive feedback.

Line 263: Remove comma after “measurement”

Edited

Line 292: Remove word “it” or add the word “is” after

Edited

Lines 343-344: This is the sentence where I suggested using something like “and provide data on animal welfare states” to end the statement rather than “can impact on animal welfare states” due to the debate around abnormal behavior and what it means for welfare.

Edited accordingly

Line 371: “then” should be “them”

Edited

Line 537: “designs” should be singular

Edited

Lines 606-607: Perhaps give an example or two of an extraneous variable in parentheses for those unfamiliar with the term.

Thank you for the suggestion. We have included two examples of extraneous variables.

Figure 6: I really like this and think it’s helpful. With introductory resources, however, I worry about some readers assuming they must do every single thing as written. Therefore, I would suggest removing reference to a meta-analysis from the figure itself as well as the caption. Done properly on species with a large amount of literature, meta-analyses are incredibly time consuming and imply the use of statistics which aren’t necessary if the researcher simply wants to review existing research and put together an ethogram of known behaviors. Meta-analyses are certainly powerful and should be used more often but they’re not necessary to begin a project. I’d simply suggest a “literature review” instead.

Thank you for the comment. We have removed the inclusion of a meta-analysis and edited the caption to explain that not all are needed to be done all at the same time, the researcher can pick and choose based on their project.

Line 752: As mixed-methods approaches can be (and frequently are) done at a single zoo, I’d recommend replacing “are” with “may be”

Edited